# BAYESIAN OPTIMIZATION BY MINIMUM FILLING DISTANCE SEARCH

## ABSTRACT

Bayesian Optimization sequentially queries objective function evaluations, often focusing on the expected utility of evaluating corresponding candidates under uncertainty with a learned probabilistic model of underlying true objective functions. We propose a new filling distance based acquisition function, termed Minimum Filling Distance Search (MFDS), to explicitly takes into account the location of the previous queried observations so that acquisition iterations can avoid oversampling and therefore explore the whole design space more efficiently. For multi-objective optimization, in addition to efficiently approaching the Pareto front, the queried candidates by MFDS are well spread over the entire Pareto set. We provide an asymptotical convergence proof and empirically evaluate MFDS performances, demonstrating the improvement over existing methods using other acquisition functions.

## 1 INTRODUCTION

Bayesian optimization (BO) is a powerful framework for optimizing unknown or uncertain objective functions, which are usually non-convex and expensive to evaluate with respect to both time and cost under limited evaluation budgets (Ahmadianshalchi et al., 2024; Wang et al., 2024; Tu et al., 2022). BO essentially resorts to sequential sampling. By incorporating uncertainty in a *probabilistic model* of unknown objective functions, on which some *acquisition functions* are built, each BO iteration optimizes the acquisition functions to evaluate the next selected query point and updates the probabilistic model accordingly with new samples. Traditional acquisition functions have been based on the posterior distribution of objective values at a single location, for example EI (Expected Improvement; Jones et al. (1998)). The information of previous queried samples is not explicitly utilized in these acquisition functions but used to iteratively update the probabilistic model. As a result, when an inappropriate prior is set for the probabilistic model, BO may fall in local optima (Wang & de Freitas, 2014). On the other hand, estimating the hyperparameters of the probabilistic model can be difficult to accomplish with very few objective function evaluations.

When multiple objectives need to be considered for the optimal design, BO can become more complicated. Instead of finding the optimum point with one objective, the goal of multi-objective BO (MOBO) is to approach the whole Pareto optimal set. Since traditionally BO selects only one query point in each iteration by optimizing the acquisition function, it is important to design the acquisition function reflecting the goodness of each sample related to the unobserved Pareto front. The dominated hypervolume indicator with respect to a reference point is adopted for Expected HyperVolume Indicator (EHVI) (Emmerich et al., 2006). However, its performance can be affected by the choice of this reference point since the hypervolume indicator can be biased towards certain regions according to the position of the reference point (Auger et al., 2009). How to specify the reference point location has been rarely studied except in Ishibuchi et al. (2017), which requires the full knowledge of the Pareto front.

Information-theoretic acquisition functions, such as entropy search (ES), aim to maximize the information gained from the next observation (Hernández-Lobato et al., 2014; 2016). Rather than directly approaching the optimal value, ES reduces the uncertainty about its location in the design space and suggests querying the most probable optimal location in the last step. This specific design works well for objective functions with a single global optimum. For objective functions in real-world applications, however, there is often not just one single optimum. What's more, in multi-objective

problems, the goal is to approach a Pareto optimal set instead of a single optimum and, therefore, the desired solution must account for the coverage of the complete Pareto front if possible. Achieving adequate coverage together with the closeness to the Pareto front necessitates the selection of acquisition points that are well distributed across regions of high potential along the Pareto front. This requires information gathering in these high-potential Pareto front regions rather than broadly reducing uncertainty across the entire design or objective space.

Our proposed acquisition is motivated by the space-filling design algorithm for batch experimental design (Pronzato & Müller, 2012). The essence is to distribute sampling inputs to cover the design space as much as possible, for example by minimizing the maximum nearest neighbor distance among the sampled inputs in the design space (minimax-distance design; Johnson et al. (1990)). In the same vein, we develop a new acquisition function for BO in particular MOBO, based on minimizing the distance between the optimal design solution, or Pareto optimal solutions, and the closest sample in the previous observation sequence. This acquisition function, by the concept of the minimum filling distance search (MFDS), enables sequential sampling concentrated in regions with a high probability of yielding optimal designs, while also exploring other high-potential areas to ensure thorough coverage of the entire design space. This mitigates the risk of oversampling trapped regions, particularly in scenarios with biased priors or a limited number of observed samples. Also, it focuses on highly likely Pareto optimal regions. Compared with previous methods, MFDS achieves superior optimal design solutions by transitioning the exploration and exploitation relationship from '*OR*' to '*AND*', establishing a more effective balance. Moreover, since MFDS aims to minimize a function of the entire sampling location sequence, it avoids the need to make a final recommendation evaluation as in ES-based methods.

The rest of the paper is organized as follows: Section 2 provides a review of relevant acquisition functions adopted in both single- and multi-objective BO. We then formulate our MFDS-based BO in Section 3 and provide the convergence proof in Section 4. With empirical results in Section 5, Section 6 concludes the paper.

## 2 BAYESIAN OPTIMIZATION

**Single-objective Bayesian optimization.** In the following single-objective BO problem:

$$\boldsymbol{x}^* = \arg\min_{\boldsymbol{x} \in \mathcal{X}} f(\boldsymbol{x}), \tag{1}$$

$\boldsymbol{x}$ is a $d$-dimensional vector of decision variables in the feasible design space $\mathcal{X} \subset \mathbb{R}^d$; and $f(\cdot) : \mathcal{X} \to \mathbb{R}$ is a continuous black-box objective function, with any evaluation $f(\boldsymbol{x})$ being an expensive process (time and/or cost). We aim to approach the global minimizer $\boldsymbol{x}^*$ with a finite function evaluation budget $N$ based on the required number of evaluations.

The prior belief of the unknown objective function $p(f)$ is described by a probabilistic model, typically a Gaussian process (GP) in BO. A GP is fully characterized by its mean function $\mu(\cdot) : \mathcal{X} \to \mathbb{R}$ and covariance kernel function $k(\cdot, \cdot) : \mathcal{X}^2 \to \mathbb{R}$. The kernel function connects the input location to the correlation of the objective values. The continuous assumption requires the correlation to converge to 1 as two points get increasingly close to each other. The convergence rate depends on the hyperparameters of the kernel function. For example, the square exponential (SE) kernel takes the form $k_{SE}(\boldsymbol{x}, \boldsymbol{x}') = \sigma_f^2 \exp\left(-\frac{(\boldsymbol{x}-\boldsymbol{x}')^T(\boldsymbol{x}-\boldsymbol{x}')}{2\sigma_l^2}\right)$ with hyperparameters $\{\sigma_f, \sigma_l\}$.

BO sequentially collects a sequence of observed samples with the corresponding evaluations. Given the observation data set at the $n$-th iteration $\mathcal{D}_n = \{\boldsymbol{X}_n, Y_n\}$, the posterior of $f$ is still a GP (Rasmussen, 2003), denoting $p(y|\boldsymbol{x}, \mathcal{D}_n) = p(f(\boldsymbol{x}) = y|\mathcal{D}_n)$.

In each iteration of BO, the next query point is chosen by optimizing an acquisition function:

$$\boldsymbol{x}_{n+1} = \arg\max_{\boldsymbol{x} \in \mathcal{X}} u_n(\boldsymbol{x}). \tag{2}$$

The acquisition function $u_n(\boldsymbol{x})$ is defined as the expected utility of evaluating $\boldsymbol{x}$ based on the updated probabilistic model. The acquisition function should balance exploitation and exploration, which means the evaluation procedure should favor both the points with potential good values with respect to the objective and the informative points from the unexplored regions for learning better surrogate models of $f$.

In most of the existing acquisition functions, the evaluation utility is tied to the objective function value. For example, the expected improvement (EI) (Jones et al., 1998) is defined as:

$$u_n^{\text{EI}}(\boldsymbol{x}) := \mathbb{E}_{y|\mathcal{D}_n, \boldsymbol{x}}[\max\{0, f(\boldsymbol{x}^+) - y\}], \tag{3}$$

where $f(\boldsymbol{x}^+) = \min_{i \leq n} f(\boldsymbol{x}_i)$ is the minimum observed value up to the $n$-th BO iteration.

In contrast, the entropy search (ES) and predictive entropy search (PES, an efficient approximation of ES) policies (Villemonteix et al., 2009; Hernández-Lobato et al., 2014) consider the posterior distribution over the unknown minimizer, denoted by $p_{\min}(\boldsymbol{x}^*|\mathcal{D}_n) = p(\boldsymbol{x}^* = \arg\min_{\boldsymbol{x}} f(\boldsymbol{x})|\mathcal{D}_n)$. ES aims to reduce the uncertainty at $\boldsymbol{x}^*$ by selecting the point $\boldsymbol{x}$ that has the largest mutual information between the function value $f(\boldsymbol{x})$ and $\boldsymbol{x}^*$. The ES acquisition function is defined as:

$$u_n^{\text{ES}}(\boldsymbol{x}) := H(\boldsymbol{x}^*|\mathcal{D}_n) - \mathbb{E}_{y|\mathcal{D}_n, \boldsymbol{x}}[H(\boldsymbol{x}^*|\mathcal{D}_n \cup \{\boldsymbol{x}, y\})], \tag{4}$$

where $H(\cdot)$ is the differential entropy function. The acquisition function of ES only focuses on reducing the uncertainty of $\boldsymbol{x}^*$, so the observation sequence selected is not necessarily close to $\boldsymbol{x}^*$. At the last iteration, ES needs to make a final suggestion of the most possible location of $\boldsymbol{x}^*$. However, when the BO problem has multiple global minimizers, it is difficult to decide how to allocate the limited evaluation budget for verifying the final suggestions since the number of global minimizers is unknown in advance.

A common limitation of these acquisition functions do not explicitly consider the location information of previously observed samples. Instead, the information is only introduced to update the surrogate GP. With the same observation history, the acquisition function will also depend on the choice of the kernel and the corresponding hyperparameter values. Inappropriate kernel and hyperparameter setups will result in sequentially observed samples cluster around some suboptimal points before covering the whole design space, as we shown in Fig. 1 when optimizing an multi-modal objective function.

**Multi-objective Bayesian optimization.** The multi-objective Bayesian optimization (MOBO) problem is to find the optimal design $\boldsymbol{x} \in \mathcal{X}$ with a set of optimal trade-off outcomes known as the *Pareto front* with respect to multiple design objectives:

$$\{f_1(\boldsymbol{x}), f_2(\boldsymbol{x}), \ldots, f_m(\boldsymbol{x})\}. \tag{5}$$

Denote the $m$-objective function image as $\mathcal{Y} \subset \mathbb{R}^m$, the Pareto front is defined as $\mathcal{Y}^* = \{\boldsymbol{y} \in \mathcal{Y} :/ \exists \boldsymbol{y}' \text{ s.t. } \boldsymbol{y}' \prec \boldsymbol{y}\}$. Here, $\boldsymbol{y} \prec \boldsymbol{y}'$ reads as $\boldsymbol{y}$ *dominates* $\boldsymbol{y}'$ in the context of minimization, meaning that $\forall i \leq m, y_i \leq y_i'$ and $\exists j \leq m, y_j < y_j'$. In the design space, the pre-image of $\mathcal{Y}^*$ is denoted by $\mathcal{X}^*$, called *Pareto optimal set*.

The sequential nature of BO still requires multi-objective BO to optimize a single acquisition function, considering all the objectives, in each iteration in order to select the next query point. That requires a single metric to describe the approximation of the observed sequence to the Pareto front. The popular acquisition function Expected HyperVolume Improvement (EHVI) uses the hypervolume indicator as the approximation metric (Emmerich et al., 2006). Assume $A \subset \mathcal{Y}$ is an objective vector set, the hypervolume indicator of $A \subset \mathcal{Y}$ is defined as the hypervolume of the dominated region $\mathcal{H}(A) = \text{Vol}(\{\boldsymbol{y} \in \mathbb{R}^m | \boldsymbol{y} \prec \boldsymbol{r} \text{ and } \exists a \in A : a \prec \boldsymbol{y}\})$. Here $\boldsymbol{r}$ is a vector dominated by all the vectors in image $\mathcal{Y}$, called the reference point, introduced to bound the objective space so that the dominated hypervolume is finite. The EHVI acquisition function is defined as:

$$u_n^{\text{EHVI}}(\boldsymbol{x}) := \mathbb{E}_{y|\mathcal{D}_n, \boldsymbol{x}}[\mathcal{H}(Y \cup y) - \mathcal{H}(Y)]. \tag{6}$$

EHVI can be biased towards certain regions, depending on the shape of the Pareto front and the position of the reference point (Auger et al., 2009). For example, if the reference point is far from the Pareto front, the EHVI will prioritize extreme or edge solutions on the Pareto front, which are typically more extreme in one or more objectives compared to the currently observed solutions. This bias occurs because edge solutions are likely to yield the greatest hypervolume improvement. However, the aim of BO should be to collect evaluation samples across the whole Pareto front to facilitate better design decision making.

As we have mentioned, the ES acquisition function can be extended to multi-objective BO, as shown in Hernández-Lobato et al. (2016) using predictive entropy search for multi-objective BO (PES-MO):

$$u^{\text{PES-MO}} := H(\mathcal{X}^*|\mathcal{D}_n) - \mathbb{E}_{y|\mathcal{D}_n, \boldsymbol{x}}[H(\mathcal{X}^*|\mathcal{D}_n \cup \{\boldsymbol{x}, y\})]. \tag{7}$$

The sampling points here only aim to reduce the uncertainty of $\mathcal{X}^*$, which may not be close to $\mathcal{X}^*$. Again, since the size of $\mathcal{X}^*$ is unknown, PES-MO cannot make good final optimal design suggestions.

To address oversampling trapped in a single region due to initial bias and suboptimal suggestions for uncertainty reduction, various strategies have been explored to balance exploration and exploitation. In *Appendix*, we discuss the advantages and limitations of these previous works in Section A.1 and provide experimental comparison results in Sections A.2 and A.3.

# 3  MINIMUM FILLING DISTANCE SEARCH

Motivated by space-filling experimental design (Pronzato & Müller, 2012), we propose a new acquisition function of the minimum filling distance search (MFDS).

**Single-objective MFDS.**  We start with the problem (1) of minimizing an unknown function $f(\boldsymbol{x})$, $\boldsymbol{x} \in \mathcal{X}$ modeled by a surrogate GP. Given an observation data set $\mathcal{D} = \{\boldsymbol{X}, Y\}$, the posterior distribution of the minimizer location is denoted as $p_{\min}(\boldsymbol{x}^*|\mathcal{D})$. We define the expected minimum distance between a sampling sequence $\boldsymbol{X}_n = \{\boldsymbol{x}_1, \boldsymbol{x}_2, \ldots, \boldsymbol{x}_n\}$ and the minimizer posterior distribution $p_{\min}(\boldsymbol{x}^*|\mathcal{D})$ as:

$$G(\boldsymbol{X}_n, \mathcal{D}) = \mathbb{E}_{\boldsymbol{x}^*|\mathcal{D}}[d_{\min}(\boldsymbol{X}_n, \boldsymbol{x}^*)] = \int_{\mathcal{X}} p_{\min}(\boldsymbol{x}^*|\mathcal{D}) d_{\min}(\boldsymbol{X}_n, \boldsymbol{x}^*) \mathrm{d}\boldsymbol{x}^*,$$

where $d_{\min}(\boldsymbol{X}_n, \boldsymbol{x}) = \min_{\boldsymbol{x}_i \in \boldsymbol{X}_n} \|\boldsymbol{x}_i - \boldsymbol{x}\|$ represents the minimal distance between the point $\boldsymbol{x}$ and the sequence $\boldsymbol{X}_n$. Note that $\boldsymbol{X}_n$ aims to capture the distribution of $p_{\min}(\boldsymbol{x}^*|\mathcal{D})$, and is not necessarily equal to the observation set $\boldsymbol{X}$. The function $G(\boldsymbol{X}_n, \mathcal{D})$ serves as a metric describing the distance between a sampling sequence $\boldsymbol{X}_n$ to the distribution of $\boldsymbol{x}^*$. A smaller value of $G$ suggests $\boldsymbol{X}_n$ is likely to be close to the unknown $\boldsymbol{x}^*$. Our proposed acquisition function is based on the $G$ function.

When we add a new observation point $\{\boldsymbol{x}, y\}$, the minimal distance between $\boldsymbol{X}_n$ and the update posterior of $\boldsymbol{x}^*$ is $G(\boldsymbol{X}_n, \mathcal{D} \cup \{\boldsymbol{x}, y\})$, average this value over the posterior of $y$ we get:

$$\mathbb{E}_{y|\boldsymbol{x}, \mathcal{D}}[G(\boldsymbol{X}_n, \mathcal{D} \cup \{\boldsymbol{x}, y\})] = \iint P(y|\boldsymbol{x}, \mathcal{D}) p_{\min}(\boldsymbol{x}^*|y, \boldsymbol{x}, \mathcal{D}) d_{\min}(\boldsymbol{X}_n, \boldsymbol{x}^*) \mathrm{d}y \mathrm{d}\boldsymbol{x}^*$$

$$= \int p_{\min}(\boldsymbol{x}^*|\mathcal{D}) d_{\min}(\boldsymbol{X}_n, \boldsymbol{x}^*) \mathrm{d}\boldsymbol{x}^* = G(\boldsymbol{X}_n, \mathcal{D}). \tag{8}$$

When any sampling point $\boldsymbol{x}$ is added to $\boldsymbol{X}_n$:

$$G(\boldsymbol{x} \cup \boldsymbol{X}_n, \mathcal{D}) \leq G(\boldsymbol{X}_n, \mathcal{D}). \tag{9}$$

These properties guarantee the convergence of MFDS as we show in Section 4.

In space-filling for batch experimental design (Johnson et al., 1990), it is to solve a min-max problem: $\min_{\boldsymbol{X}_n} \sup_{\boldsymbol{x} \in \mathcal{X}} d_{\min}(\boldsymbol{X}_n, \boldsymbol{x})$ so that the design sample points can be uniformly spread over the design space. In this paper, we aim to sequentially sample points close to the unknown minimizer point, so our objective function $G$ averages over the minimizer location distribution, taking into account the uncertainty from the surrogate GP based on observed samples. More precisely, let $N$ be the total budget of evaluations, the goal of our problem is to choose an evaluation policy $\pi \in \Pi$ to derive the sampling sequence $\boldsymbol{X}_N$ minimizing the expected minimum distance:

$$\inf_{\pi \in \Pi} \mathbb{E}^\pi[G(\boldsymbol{X}_N, \mathcal{D}_N)]. \tag{10}$$

With this objective, the sampling points will be dense in regions where the posterior distribution $p_{\min}(\boldsymbol{x}^*|\boldsymbol{X}_N, Y_N)$ is high, making $d_{\min}(\boldsymbol{X}_N, \boldsymbol{x}^*)$ small in these regions, while sparse sampling points will be places in regions with lower $p_{\min}(\boldsymbol{x}^*|\boldsymbol{X}_N, Y_N)$, allowing for exploration in less-explored regions. Therefore, the method balances exploitation of knowledge about the predicted minimizer with exploration of under-sampled regions.

The sequential decision problem equation 10 can be expressed and solved with a dynamic programming formulation, which inspires our acquisition function. The state is the observation set $\mathcal{D}_n$, and the policy $\pi \in \Pi$ is defined as $x_{n+1} = \pi(\mathcal{D}_n)$. Define the value function at iteration $n$ as:

$$V_n(\mathcal{D}_n) = \inf_{\pi \in \Pi} \mathbb{E}^\pi[G(\boldsymbol{X}_N, \mathcal{D}_N)|\mathcal{D}_n] \tag{11}$$

When $n = N$, the value function is just $V_N(\mathcal{D}_N) = G(\boldsymbol{X}_N, \mathcal{D}_N)$. Based on the Bellman equation (Bertsekas et al., 1995), the value function at $0 \leq n < N$ can be calculated recursively by

$$V_n(\mathcal{D}_n) = \min_{\boldsymbol{x}} \mathbb{E}_{y|\boldsymbol{x}, \mathcal{D}_n}[V_{n+1}(\mathcal{D}_n \cup \{\boldsymbol{x}, y\})]. \tag{12}$$

Specially, for $n = N - 1$, the optimal policy is just a greedy method:

$$V_{N-1}(\mathcal{D}_{N-1}) = \min_{\boldsymbol{x}} \mathbb{E}_{y|\boldsymbol{x}, \mathcal{D}_{N-1}}[G(\boldsymbol{x} \cup \boldsymbol{X}_{N-1}, \mathcal{D}_{N-1} \cup \{\boldsymbol{x}, y\})] = \min_{\boldsymbol{x}} G(\boldsymbol{x} \cup \boldsymbol{X}_{N-1}, \mathcal{D}_{N-1}).$$

For $n = N - 2$, the value function is related to the one-step-look-ahead policy (Novoa & Storer, 2009):

$$\begin{aligned}
V_{N-2}(\mathcal{D}_{N-2}) &= \min_{\boldsymbol{x}} \mathbb{E}_{y|\boldsymbol{x}, \mathcal{D}_{N-2}}[V_{N-1}(\mathcal{D}_{N-2} \cup \{\boldsymbol{x}, y\})] \\
&= \min_{\boldsymbol{x}} \mathbb{E}_{y|\boldsymbol{x}, \mathcal{D}_{N-2}}[\min_{\boldsymbol{x}''} G(\{\boldsymbol{x}, \boldsymbol{x}''\} \cup \boldsymbol{X}_{N-2}, \mathcal{D}_{N-2} \cup \{\boldsymbol{x}, y\}))]. \tag{13}
\end{aligned}$$

We can use one-step-look-ahead (OSLA) methods to approximate the optimal policy (Novoa & Storer, 2009), with the acquisition function considering both long-term and short-term rewards:

$$u_n(\boldsymbol{x}) = \min_{\boldsymbol{x}'} G(\boldsymbol{x}' \cup \boldsymbol{X}_n, \mathcal{D}_n) - \mathbb{E}_{y|\boldsymbol{x}, \mathcal{D}_n}[\min_{\boldsymbol{x}''} G(\{\boldsymbol{x}, \boldsymbol{x}''\} \cup \boldsymbol{X}_n, \mathcal{D}_n \cup \{\boldsymbol{x}, y\})].$$

The first term is the result of the greedy policy to reduce the filling distance and the second term is the one-step-look-ahead policy. The acquisition function can be interpreted as the benefit brought about by reducing the uncertainty at point $\boldsymbol{x}$. In each iteration, we take the maximum point of the acquisition function as the next observation sample. After $N - 1$ iterations, the algorithm makes a greedy evaluation at time $N$ with $\boldsymbol{x}_N = \arg\max_{\boldsymbol{x}} G(\boldsymbol{x} \cup \boldsymbol{X}_{N-1}, \mathcal{D}_{N-1})$. In the following section, we'll prove that following this procedure, as $N \to \infty$, the minimizer point will be an adherent point of the observation sequence with probability 1.

Compared with existing acquisition functions shown in the Section A.1, MFDS aims to minimize a utility function of the whole sampling location sequence, so it can avoid sampling points clustering together. In contrast to ES providing a final suggestion of the most possible minimizer location, MFDS provides an observation sequence that covers all the possible region of the minimizer.

**Multi-objective MFDS.** We now turn to multi-objective BO to approach the Pareto-optimal set $\mathcal{X}^* \subseteq \mathcal{X}$. Besides requiring that the observed samples be close to the Pareto-optimal set, it is also crucial to consider how well these observed points cover its entire range. Ideally, the observed points should be distributed as uniformly as possible along on the Pareto-optimal set. The average coverage quality of the observed set $\boldsymbol{X}_n$ on the Pareto manifold $\mathcal{X}^*$ can be defined as:

$$d_{\min}^*(\boldsymbol{X}_n, \mathcal{X}^*) = \frac{\int_{\mathcal{X}^*} d_{\min}(\boldsymbol{X}_n, \boldsymbol{x}^*) \mathrm{d}\mu(\boldsymbol{x}^*)}{\mu(\mathcal{X}^*)}, \tag{14}$$

where $\mu(\mathcal{X}^*) = \int_{\mathcal{X}^*} d\mu(\boldsymbol{x}^*)$ is the Riemannian volume with the Riemannian measure $\mu(\cdot)$. Define $\Omega$ as the collection of surrogate realizations. For each random realization $\omega \in \Omega$, the Pareto front is $\mathcal{X}^*(\omega)$. The extended $G$ function for multi-objective MFDS can be formulated as:

$$\begin{aligned}
G^*(\boldsymbol{X}_n, \mathcal{D}) &= \mathbb{E}_{\mathcal{X}^*|\mathcal{D}}[d_{\min}^*(\boldsymbol{X}_n, \mathcal{X}^*)] = \int_{\Omega} P(\mathcal{X}^*(\omega)|\mathcal{D}) \frac{\int_{\mathcal{X}^*(\omega)} d_{\min}(\boldsymbol{X}_n, \boldsymbol{x}) \mathrm{d}\mu_{\omega}(\boldsymbol{x})}{\mu_{\omega}(\mathcal{X}^*(\omega))} \mathrm{d}\omega \\
&= \int_{\mathcal{X}} \left( \int_{\Omega} \frac{\mathbb{1}(\boldsymbol{x} \in \mathcal{X}^*(\omega)) P(\mathcal{X}^*(\omega)|\mathcal{D})}{\mathrm{Vol}(\mathcal{X}^*(\omega))} \mathrm{d}\omega \right) d_{\min}(\boldsymbol{X}_n, \boldsymbol{x}) \mathrm{d}\mu(\boldsymbol{x}) \\
&= \int_{\mathcal{X}} p_{\min}^*(\boldsymbol{x}|\mathcal{D}) d_{\min}(\boldsymbol{X}_n, \boldsymbol{x}) \mathrm{d}\mu(\boldsymbol{x}), \tag{15}
\end{aligned}$$

where $\mathbb{1}(\boldsymbol{x} \in \mathcal{X}^*(\omega))$ is the indicator of whether $\boldsymbol{x}$ lies on the manifold $\mathcal{X}^*(\omega)$. We define

$$p_{\min}^*(\boldsymbol{x}|\mathcal{D}) := \int_{\Omega} \frac{\mathbb{1}(\boldsymbol{x} \in \mathcal{X}^*(\omega)) P(\mathcal{X}^*(\omega)|\mathcal{D})}{\mu_{\omega}(\mathcal{X}^*(\omega))} \mathrm{d}\omega \tag{16}$$

as the average probability distribution of $\boldsymbol{x} \in \mathcal{X}^*$. It is easy to verify that $\int_{\mathcal{X}} p_{\min}^*(\boldsymbol{x}|\mathcal{D}) \mathrm{d}\boldsymbol{x} = 1$, so that the properties of $G$ function still hold for $G^*$.

Based on the $G^*$ function extended for multi-objective problems, the acquisition function for multi-objective MFDS is built in the same way:

$$u_n^*(\boldsymbol{x}) = \min_{\boldsymbol{x}'} G^*(\boldsymbol{x}' \cup \boldsymbol{X}_n, \mathcal{D}_n) - \mathbb{E}_{y|\boldsymbol{x}, \mathcal{D}_n} \left[ \min_{\boldsymbol{x}''} G^*\left(\{\boldsymbol{x}, \boldsymbol{x}''\} \cup \boldsymbol{X}_n, \mathcal{D}_n \cup \{\boldsymbol{x}, y\}\right)\right]. \quad (17)$$

As we directly extend the acquisition function for single-objective MFDS above to multi-objective cases, there is no need for additional heuristics, including choosing the reference point as in EHVI-based methods. For multi-objective MFDS, the sampling procedure, which is the same as the one for single-objective BO solutions, optimizes (17) in the first $N - 1$ iterations and makes a greedy evaluation at the last iteration.

## 4 CONVERGENCE PROOF FOR MFDS

We now present the theoretical guarantees for our single-objective MFDS-based BO procedure. When optimizing (17) in the first $N - 1$ iterations and performing a greedy evaluation in the final iteration, the collected sampling sequence $\boldsymbol{X}_N$ will almost surely include $\boldsymbol{x}^*$ as an adherent point as $N \to \infty$. Before delving into the detailed proofs, we outline the critical steps for the proof, which resembles the logic flow of the convergence proof for EI-based BO in Vazquez & Bect (2010): (1) we first prove that any sampling location sequence $\boldsymbol{X}_n$ in the bounded design space must satisfy $u_n(\boldsymbol{x}_{n+1}) \to 0$; (2) since $u_n(\boldsymbol{x}_{n+1})$ is the maximum of the acquisition function over the design space, we conclude that the acquisition function uniformly converges to 0; (3) we finally show that $\min_{x'} G(x' \cup \boldsymbol{X}_n, \mathcal{D}_n) \to 0$ almost surely. The theoretical analysis of MFDS-based multi-objective BO is left for future work.

For the following proofs, we assume that the objective function $f(\cdot)$ is an element of a Reproducing Kernel Hilbert Space (RKHS) of kernel $k(\cdot, \cdot)$, which is a continuous and positive definite kernel function. Denote the posterior mean of a GP at $\boldsymbol{x}$ as $\mu(\boldsymbol{x}|\mathcal{D}_n)$ and the variance as $\sigma^2(\boldsymbol{x}|\mathcal{D}_n)$, we first define a property for the kernel function.

**Definition 1.** *Assume that the kernel function $k(\cdot, \cdot)$ is continuous and positive definite. We say that $k(\cdot, \cdot)$ has the Smooth Minimizer Distribution (SMD) property if for any location sequence $\boldsymbol{X}_n \in \mathcal{X}^n$, $\boldsymbol{x} \in \mathcal{X}$ and $y \in \mathbb{R}$, the following assertion is true: As $d_{min}(\boldsymbol{x}, \boldsymbol{X}_n) \to 0$, we have:*

$$p_{min}(\boldsymbol{x}^*|\mathcal{D}_n \cup \{\boldsymbol{x}, \mu(\boldsymbol{x}|\mathcal{D}_n)\}) \xrightarrow{unif.} p_{min}(\boldsymbol{x}^*|\mathcal{D}_n). \quad (18)$$

As proved in **Proposition 10** in Vazquez & Bect (2010), if $\boldsymbol{x}$ is an adherent point of $\boldsymbol{X}_n$, then $\mu(\boldsymbol{x}|\mathcal{D}_n)$ will converge to the real function value $f(\boldsymbol{x})$. So the SMD property actually states that one additional observation that is close to the observed dataset $\mathcal{D}_n$ won't change much on the distribution $p_{\min}(\boldsymbol{x}^*|\mathcal{D}_n)$. Since the theory of suprema of stochastic processes is nontrivial, a necessary and sufficient condition for the SMD property is an open problem, to the best of our knowledge.

**Theorem 1.** *Assume that the design space $\mathcal{X} \subset \mathbb{R}^d$ is compact and bounded for some $d \leq 1$, the objective function $f(\cdot)$ is an element of a Reproducing Kernel Hilbert Space (RKHS) of kernel $k(\cdot, \cdot)$, which is a continuous and positive definite kernel function and has the SMD property. The observation dataset $\mathcal{D}_n$ generated by our MFDS-based BO has the following property: as $n \to \infty$, $\min_{x'} G(x' \cup \boldsymbol{X}_n, \mathcal{D}_n) \to 0$.*

First, we prove the following lemma for the infimum of the MFDS utility function:

**Lemma 1.** $u_n(\boldsymbol{x}) \geq 0, \forall \boldsymbol{x} \in \mathcal{X}$

**Proof.** Assume $\boldsymbol{x}' = \arg\min_{\boldsymbol{x}} G(\boldsymbol{x} \cup \boldsymbol{X}_n, \mathcal{D}_n)$ is the minimizer of the first term in (17) . The second term of (17) can be bounded as follows:

$$\mathbb{E}_{y|\boldsymbol{x}, \mathcal{D}_n}[\min_{\boldsymbol{x}''} G(\{\boldsymbol{x}, \boldsymbol{x}''\} \cup \boldsymbol{X}_n, \mathcal{D}_n \cup \{\boldsymbol{x}, y\})] \quad (19)$$

$$\leq \mathbb{E}_{y|\boldsymbol{x}, \mathcal{D}_n}[G(\{\boldsymbol{x}, \boldsymbol{x}'\} \cup \boldsymbol{X}_n, \mathcal{D}_n \cup \{\boldsymbol{x}, y\})] = G(\{\boldsymbol{x}, \boldsymbol{x}'\} \cup \boldsymbol{X}_n, \mathcal{D}_n) \leq G(\boldsymbol{x}' \cup \boldsymbol{X}_n, \mathcal{D}_n);$$

hence $u_n(\boldsymbol{x}) \geq 0, \forall \boldsymbol{x} \in \mathcal{X}$. This concludes the proof. $\square$

**Lemma 2.** *If $k(\cdot, \cdot)$ is continuous and positive definite and has the SMD property, $\boldsymbol{x}$ is an adherent point of sequence $\boldsymbol{X}_n$, i.e. $\lim_{n \to \infty} d_{min}(\boldsymbol{x}, \boldsymbol{X}_n) = 0$, then $\lim_{n \to \infty} u_n(\boldsymbol{x}) = 0$.*

**Proof.** If $\boldsymbol{x} \in \boldsymbol{X}_n$, the result holds trivially. Assume that $\boldsymbol{x} \notin \boldsymbol{X}_n$. As proved in **Proposition 10** in Vazquez & Bect (2010), since $\boldsymbol{x}$ is an adherent point of $\boldsymbol{X}_n$ and the kernel function $k$ is continuous, the variance $\sigma^2(\boldsymbol{x}|\mathcal{D}_n) \to 0$. Then the distribution $p(y|\boldsymbol{x}, \mathcal{D}_n) = \mathcal{N}(\mu(\boldsymbol{x}|\mathcal{D}_n), \sigma^2(\boldsymbol{x}|\mathcal{D}_n))$ pointwisely converges to the Dirac delta function $\delta(y - \mu(\boldsymbol{x}|\mathcal{D}_n))$. According to the property of the delta function, the second term of (17) :

$$\mathbb{E}_{y|\boldsymbol{x},\mathcal{D}_n}[\min_{\boldsymbol{x}''} G(\{\boldsymbol{x}, \boldsymbol{x}''\} \cup \boldsymbol{X}_n, \mathcal{D}_n \cup \{\boldsymbol{x}, y\})]$$

$$\to \min_{\boldsymbol{x}''} G(\{\boldsymbol{x}, \boldsymbol{x}''\} \cup \boldsymbol{X}_n, \mathcal{D}_n \cup \{\boldsymbol{x}, \mu(\boldsymbol{x}|\mathcal{D}_n)\}) = \min_{\boldsymbol{x}''} \mathbb{E}_{\boldsymbol{x}^*|\mathcal{D}_n, \boldsymbol{x}, \mu(\boldsymbol{x}|\mathcal{D}_n)}[d_{\min}(\boldsymbol{x}^*, \boldsymbol{X}_n \cup \{\boldsymbol{x}, \boldsymbol{x}''\})]$$

The SMD property indicates that $p_{\min}(\boldsymbol{x}^*|\mathcal{D}_n \cup \{\boldsymbol{x}, \mu(\boldsymbol{x}|\mathcal{D}_n)\})$ uniformly converges to $p_{\min}(\boldsymbol{x}^*|\mathcal{D}_n)$. $\boldsymbol{x}$ is an adherent point of $\boldsymbol{X}_n$ indicates that $d_{\min}(\boldsymbol{x}^*, \boldsymbol{X}_n \cup \{\boldsymbol{x}, \boldsymbol{x}''\})$ converge to $d_{\min}(\boldsymbol{x}^*, \boldsymbol{X}_n \cup \{\boldsymbol{x}''\})$. We can therefore conclude:

$$\mathbb{E}_{y|\boldsymbol{x},\mathcal{D}_n}[\min_{\boldsymbol{x}''} G(\{\boldsymbol{x}, \boldsymbol{x}''\} \cup \boldsymbol{X}_n, \mathcal{D}_n \cup \{\boldsymbol{x}, y\})]$$

$$\to \min_{\boldsymbol{x}''} \mathbb{E}_{\boldsymbol{x}^*|\mathcal{D}_n}[d_{\min}(\boldsymbol{x}^*, \boldsymbol{X}_n \cup \{\boldsymbol{x}''\})] = \min_{\boldsymbol{x}''} G(\boldsymbol{x}'' \cup \boldsymbol{X}_n, \mathcal{D}_n), \tag{20}$$

which is just the first term of (17) and hence proves the lemma. $\square$

Define $\nu_n = \max_{\boldsymbol{x} \in \mathcal{X}} u_n(\boldsymbol{x})$, the following lemma states the asymptotic property of $\nu_n$.

**Lemma 3.** *With BO using MFDS, as $n \to \infty$, we have $\lim_{n\to\infty} \nu_n = 0$.*

**Proof.** Since $\mathcal{X}$ is bounded, elements of the infinity location sequence $\boldsymbol{X}_{n\to\infty}$ must fall into the neighborhood of previous elements. So we have $\lim_{n\to\infty} d_{\min}(\boldsymbol{X}_n, \boldsymbol{x}_{n+1}) = 0$. Note that with our MFDS-based BO algorithm, we will evaluate the maximizer of $u_n(\boldsymbol{x})$ at the $(n+1)$-th iteration, i.e. $\nu_n = u_n(\boldsymbol{x}_{n+1})$. From **Lemma 2**, we can get $\lim_{n\to\infty} \nu_n = 0$. $\square$

The following lemma is critical for the proof of **Theorem 1**:

**Lemma 4.** *At $n$-th iteration, if $\forall \boldsymbol{x} \in \mathcal{X}, u_n(\boldsymbol{x}) = 0$, then $\min_{\boldsymbol{x}'} G(\boldsymbol{x}' \cup \boldsymbol{X}_n, \mathcal{D}_n) = 0$*

A detailed proof of Lemma 4 is given in *Appendix* A.4. From **Lemma 1**, we know $\nu_n = 0$ indicates $\forall x \in \mathcal{X}, u_n(\boldsymbol{x}) = 0$. Combining this with **Lemma 3** and **Lemma 4**, we can prove that as $n \to \infty$, $\lim_{n\to\infty} G(x' \cup X_n, \mathcal{D}_n) = 0$, completing the proof for **Theorem 1**.

# 5 EXPERIMENTS

We expect that our MFDS acquisition function better balances exploitation and exploration for more efficient BO, especially when the unknown objective functions are complicated with more than one potential optimal design solutions. We have shown theoretically that single-objective MFDS will be able to identify all the optimal locations given a large enough budget. In this section, we first give an illustrative example comparing MFDS with EI and PES. We then apply MFDS to both single-objective and multi-objective problems to demonstrate its advantages over existing BO methods. To distinguish the benefits of MFDS and OSLA, we denote the greedy strategy simply as *MFDS*, while the main strategy we propose is labeled as *MFDS(OSLA)*.

## 5.1 SINGLE-OBJECTIVE OPTIMIZATION

This first single-objective BO example is to optimize $f_{GM}(x) = -0.5\mathcal{N}(-\mu, 2\sigma^2) - 0.55\mathcal{N}(\mu, \sigma^2) + 1$ with $\mu = 0.3$ and $\sigma = 0.05$. The objective function has one local minimum and one global minimum far from each other. For all the algorithms, a GP with the SE kernel is taken as the surrogate model, with the hyperparameters set to $\sigma_l = 0.01$, $\sigma_f = 1$ throughout the BO procedure. Small $\sigma_l$ here simulates the situation in which an observation point can only provide objective value information for points nearby. As a result BO with some acquisition function may only sample points close to each other. We apply BO using EI, PES and MFDS acquisition functions with the same initial set. The performance comparison is shown in Fig. 1. EI and PES fail to find the global minimum within 10 iterations, while MFDS can find the global minimum. As illustrated, EI gets trapped in the local minimum. PES also fails to identify the global minimum even it gets close to it, suffering from its focus on information gain instead of exploitation in potential optimal region. MFDS works effectively because it can achieve rapid exploitation in high-potential optimal

region, and efficient global exploration. Traditionally, these two properties have been addressed separately, as exemplified by UCB and $\epsilon$-EI. Exploration strategies typically target regions with high uncertainty or employ random sampling, without explicitly considering their potential to contain the optimum. These strategies may result in redundant evaluations, particularly under limited budgets. MFDS integrates these two properties by the integral of $p^*_{\min}(\boldsymbol{x}|\mathcal{D})d_{\min}(\boldsymbol{X}_n, \boldsymbol{x})$. This formulation ensures that MFDS prioritizes space-filling within high-potential regions. As Fig. 1c shows, once the current region is adequately filled, term $d_{\min}(\boldsymbol{X}_n, \boldsymbol{x})$ drops to small value and MFDS transits to the next promising region where $p^*_{\min}(\boldsymbol{x}|\mathcal{D})$ and $d_{\min}(\boldsymbol{X}_n, \boldsymbol{x})$ are both high.

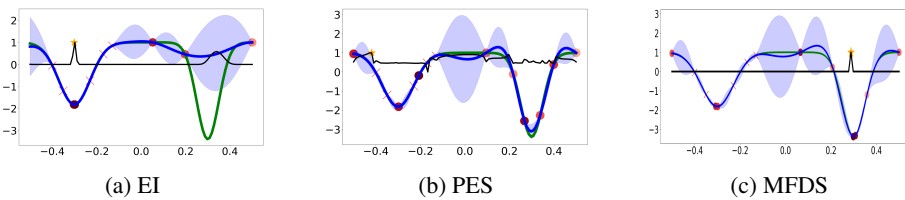

| (a) EI | (b) PES | (c) MFDS |
|---|---|---|

Figure 1: Performance comparison between EI-, PES- and MFDS-based BO. Red crosses: initial points; Red dots: Sampling points. Color darkens as iteration count increases; Blue lines: surrogate mean; Shadow areas: $95\%$ confidence interval; Yellow stars: Next acquisition; Black line: Acquisition function; Green line: Objective function.

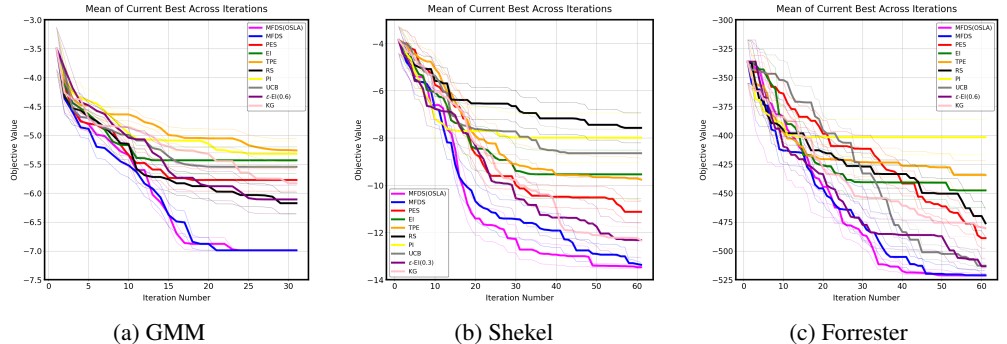

| (a) GMM | (b) Shekel | (c) Forrester |
|---|---|---|

Figure 2: Performance comparison for SOBO with different acquisition functions: (a)$f_{GM}$; (b) Shekel and (c) Forrester.

Beyond the BO example on function $f_{GM}$, we further evaluate MFDS on the Shekel and Forrester functions, both of which are multimodal. The results are summarized in Fig. 2. As shown, MFDS and its one-step-look-ahead variant consistently achieve superior mean performance relative to other methods. While MFDS can occasionally be outperformed by baselines such as PI or $\epsilon$-EI at early iterations, its advantage becomes clear as the optimization progresses: once the acquisition reaches the global optimum, MFDS reliably outperforms others by avoiding oversampling and inefficient exploration. The smaller standard deviation further shows that MFDS is not only more effective but also more stable.

## 5.2 MULTI-OBJECTIVE OPTIMIZATION

We now evaluate the MOBO performance of an acquisition function in two aspects: 1) *Hypervolume* with the reference point at the origin, and 2) Pareto front *coverage*.

*Inverted Generational Distance* (IGD; Coello Coello & Reyes Sierra (2004)) has been used to evaluate Pareto front coverage by measuring the minimum distance between the suggested candidates and the ground-truth Pareto front in the objective space. However, this metric significantly correlates with hypervolume, as candidates achieving a large hypervolume naturally exhibit short distances to the Pareto front, and vice versa. To clearly distinguish between coverage and hypervolume, we introduce a new metric, *average minimum distance* (AMD), defined in equation 14 quantifying the proximity between the selected candidates and the Pareto front set in the design space. AMD provides a more direct and independent evaluation of how well the search explores the design space, ensuring comprehensive Pareto front coverage.

To demonstrate the performance of MFDS in MOBO, we first adopt a bi-objective Gaussian mixture model (GMM) (Reynolds et al., 2009; Daulton et al., 2022) in a two-dimensional design space. Implementation details of different acquisition functions can be found in Section A.1. We report the average results over 20 trials with each running 30 iterations. Fig. 3a compares the average of the best hypervolume values. Figs. 3e and 3i illustrate the corresponding AMD and IGD at each iteration. MFDS consistently outperforms the other methods in both hypervolume and Pareto-optimal set coverage. The IGD value by MFDS also reaches a competitive level. The error bars represent the standard deviation of the corresponding evaluation metrics over 20 trials, illustrating the robustness of MOBO with different acquisition functions under varying initial conditions. In this GMM-based bi-objective example, MFDS not only achieves superior Pareto hypervolume and coverage performances but also exhibits lower variance compared to other methods. This suggests that MFDS offers better stability in optimization across different scenarios.

For other methods, EHVI initially performs well, particularly in the first five iterations, where it demonstrates strong local exploitation. However, after the local optimum is identified, EHVI continues acquiring points in the same region, resulting in diminished improvements in hypervolume and coverage, as shown in Figs 5 and 6. PES-MO demonstrates stronger exploration capabilities than EHVI, but as previously noted, PES-MO focuses on reducing uncertainty around the global optimizer and lacks the capacity to recommend a final optimal solution compared to MFDS. This disadvantage of PES-MO is observed in Figs 4 and 6. TPE-MO, while efficient, is highly dependent on the initial sampling distribution. When initial points are far from the global optima, additional iterations may be required for convergence. Both hyperparameter tuning based methods, MGF and $\epsilon$-EHVI, demonstrate strong performance. The optimal hyperparameters of both methods are selected from the range $(0, 1)$ with a step size of $0.1$. However, identifying the optimal hyperparameters requires additional trials across the entire hyperparameter search space. Even with these additional objective evaluations, their performance remains inferior to that of MFDS. The primary limitation of these hyperparameter tuning methods is that pre-set or schedule-based hyperparameters cannot adapt dynamically to the current feedback. EMMI shows similar performance compared with MGF and $\epsilon$-EHVI. However, as illustrated in Figs. 3e and 3i, coverage performance remains limited since EMMI relies on the surrogate model that introduces bias toward specific regions, influenced by initialization.

Section A.2 in *Appendix* provides further insight into search behaviors of MOBO methods using different acquisition functions. We also conduct experiments on standard MOBO benchmark functions, including DTLZ2, and provide robustness analysis against noise in Section A.3. The results demonstrate that MFDS exhibits higher robustness across varying noise levels compared to other methods.

## 5.3 REAL-WORLD APPLICATIONS

We further implement our MFDS-based MOBO and compare with other MOBO methods based on different acquisition functions on multiple real-case problems: RE2-4-1, the four-bar truss design problem (Cheng & Li, 1999), which aims to minimize structural volume and joint displacement in a four-dimensional input space; RE3-5-4, the vehicle crashworthiness design problem (Tanabe & Ishibuchi, 2020), which seeks to minimize weight, acceleration characteristics, and toe-board intrusion using five input variables; and RE3-7-5, the speed reducer design problem (Farhang-Mehr & Azarm, 2002), which minimizes volume, shaft stress, and the sum of 11 constraint violations in a seven-dimensional input space. The reference point for EHVI is set at 1.1 times the worst objective value for each objective, as derived from the approximate front provided by Tanabe & Ishibuchi (2020). We note that, since the best-case hypervolume computation from Tanabe & Ishibuchi (2020) relies on Pareto front estimation, the true Pareto hypervolume remains unknown. Our primary goal is to evaluate whether different acquisition functions can yield a Pareto set recommendation that closely aligns with the ideal Pareto optimal set.

As illustrated in Figs. 3b, 3c, 3d and 3g, our MFDS again demonstrates superior performance in both Pareto hypervolume and design space coverage evaluation metrics compared to the other acquisition functions. In contrast, the performance of EHVI significantly declines after the first five iterations, a trend also observed in the GMM example presented in Section 5.2. The coverage performance of EHVI similarly deteriorates, ultimately performing worse than RS after 20 iterations. While PES-MO maintains robust exploration capabilities across the complex feasible region

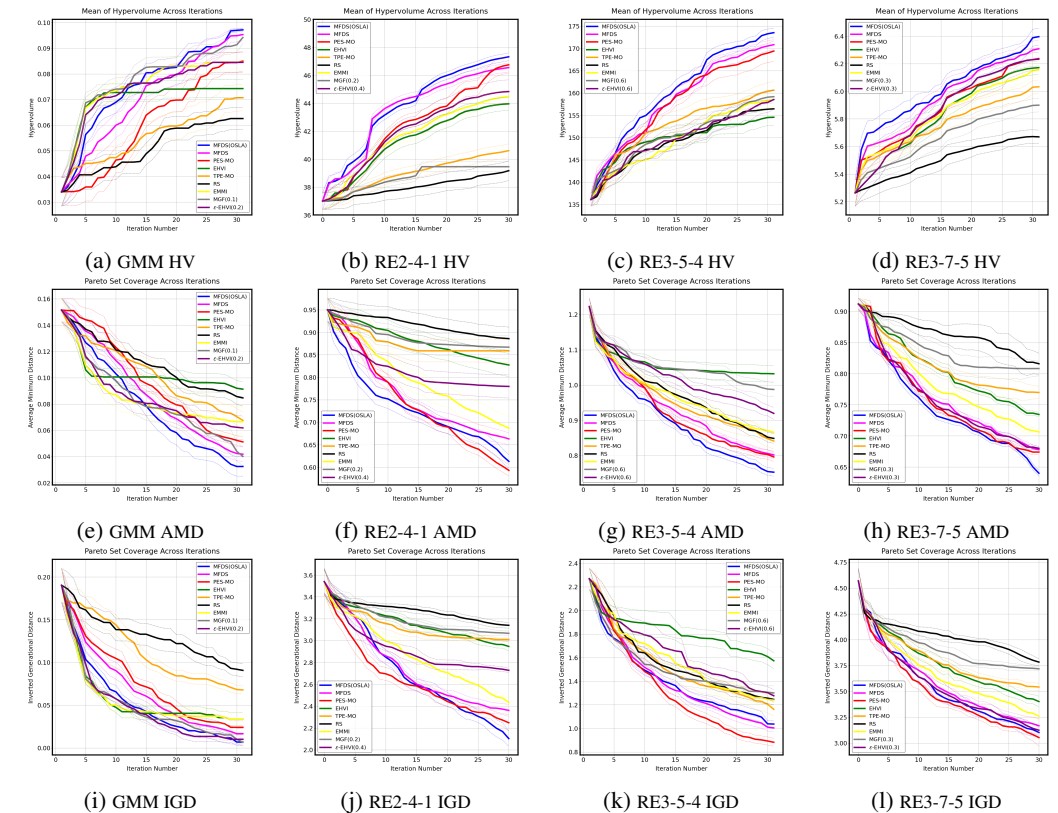

Figure 3: Performance comparison for MOBO with different acquisition functions.

as Figs. 3f, 3k and 3l shows, but it lacks the exploitation efficiency demonstrated by MFDS as illustrated in Figs. 3b, 3c and 3d. Notably, TPE-MO shows improved EHVI performance in escaping local optima, attributed to its robustness when navigating more intricate landscapes. Additional performance metrics, including standard deviation details in hypervolume and AMD, can be found in Section A.5 of *Appendix*. For the hyperparameter tuning based methods, MGF and $\epsilon$-EHVI, the tuning strategy follows the approach described in Section 5.2. However, unlike the strong performance observed in Section 5.2, these tuning-based methods fail to match the performance of MFDS. Fixed or scheduled tuning strategies for balancing exploration and exploitation are outperformed by MFDS's dynamic adaptation, particularly when optimizing objective functions with more complex landscapes. Besides the performance analysis, we also provide run-time comparison with experimental settings in Section A.6.

## 6 CONCLUSIONS & FUTURE RESEARCH

We have proposed a novel acquisition function, MFDS, for BO, considering the expected minimum distance between the sampling locations and the unknown optimal points under uncertainty. The acquisition function utilizing the design space information distinguishes our method from existing BO methods, allowing for better balance between exploitation and exploration. MFDS can effectively cover Pareto fronts or multiple optimal solutions. We prove the convergence of our method, and demonstrate superior empirical performances in our experiments.

One of future research directions is to address the computational complexity of MFDS, which is currently optimized by sampling. Similar to ES (Hennig & Schuler, 2012), the computational cost of MFDS is $O(SN^3)$, where $S$ represents the number of Monte Carlo samples and $N^3$ results from retraining the GP for each sample. As $N$ increases, the evaluation time becomes prohibitively high. To mitigate this, we plan to explore smooth approximations of MFDS to enable gradient-based optimization, allowing more efficient local search (Zhao et al., 2021).

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

# A APPENDIX

## A.1 RELATED WORKS

To mitigate oversampling in a single region due to initial bias and suboptimal suggestions from uncertainty reduction, researchers have explored hybrid approaches, such as UCB (Srinivas et al., 2009) and MGF (Moment Generating Function) (Wang et al., 2017), aim to balance exploration and exploitation with hyperparameters that guide acquisitions toward regions of high uncertainty when the process gets trapped in local optima. These methods typically favor either exploitation-driven or uncertainty-reduction-driven decisions in specific iterations, but rarely balance both simultaneously. Alternative hybrid methods, such as $\epsilon$-EI, enforce exploration in random iterations with a fixed probability $\epsilon$. Researchers also design schedules that decrease exploration as iterations progress. However, the optimal balance between exploration and exploitation is problem-dependent, making it challenging to predefine a universally effective schedule for diverse unknown objectives. To address this challenge, online tuning strategies are employed to dynamically adjust hyperparameters or schedules. Unfortunately, these strategies usually require at least 50 to 100 iterations to converge to optimal settings and can be even more computationally expensive for problems of high complexity.

Space-filling in the objective space has been adopted to improve Pareto front coverage in Expected Maximin Improvement (EMMI; Olofsson et al. (2018)). Unlike EHVI, which prioritizes improvement in the best-performing regions, EMMI emphasizes improvement in unexplored objective-space regions, aiming to enhance Pareto front coverage. However, implementing space-filling in the objective space presents several challenges: 1) Validity: It is unclear whether any candidates actually exist in the regions of the objective space being targeted for filling; 2) Bias: For black-box objective functions, the approach depends on surrogate models for estimation. This reliance introduces the risk of bias, similar to the challenge faced by EHVI, where the indicator may favor specific regions based on initialization; 3) Misalignment with Optimization Goals: Space-filling strategies may suggest queries that do not align with the hypervolume improvement direction, potentially reducing the overall optimization performance. We note that our MFDS is in the design space, which avoids potential issues from 1) and 3) and also reducing the impact by 2).

## A.2 SEARCHING BEHAVIORS OF DIFFERENT MOBO METHODS FOR THE GMM EXAMPLE

In this section, we illustrate the sampled points by MOBO methods with different acquisition functions in both the design and objective spaces for the GMM example. Figs. 4 and 5 illustrate how different MOBO methods approach the Pareto optimal set in the design space. In Fig. 6, the iterative searching behaviors of different MOBO methods are visualized with respect to the Pareto front points in the bi-objective space.

Our proposed MOBO with MFDS shows the best coverage of the Pareto front points reaching all of the Pareto front points. As we discussed in the main text, EHVI gets stuck at the left bottom corner after identifying a local subset of the Pareto front points due to its inherent bias. PES-MO demonstrates strong exploratory capabilities in covering the Pareto front. However, it shows limitations in effectively acquiring final Pareto-optimal points. Once it approaches the near-Pareto region, the acquired samples tend to drift away from this area, resulting in insufficient exploitation to fully refine the solution set. For TPE-MO, being an acquisition function based on EI, its behavior is similar as EHVI. It tends to focus on the Pareto subset in the bottom left region. However, the incorporation of a linear forgetting weight factor in TPE-MO for earlier acquired samples (Bergstra et al., 2013) partially addresses the issues of getting trapped this local optimal region by reducing their influence on density estimation for TPE.

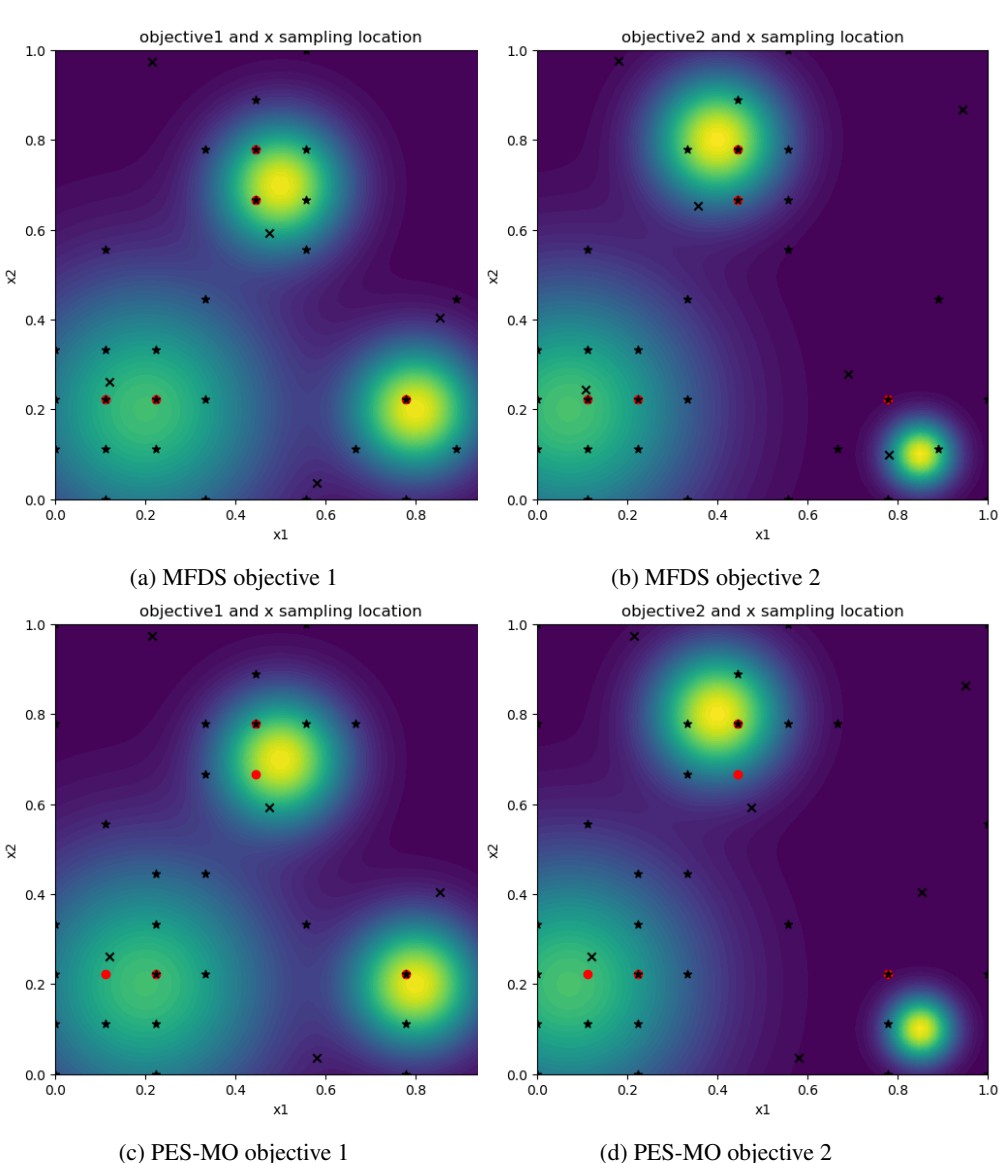

Figure 4: Searching behavior comparison in the design space for MOBO with MFDS and PES-MO. Red points indicate the Pareto set. Black stars represent the samples selected by MOBO with the corresponding acquisition functions. Background is painted with the viridis colormap. Brighter regions have larger values and darker otherwise. Crosses are the initial data.

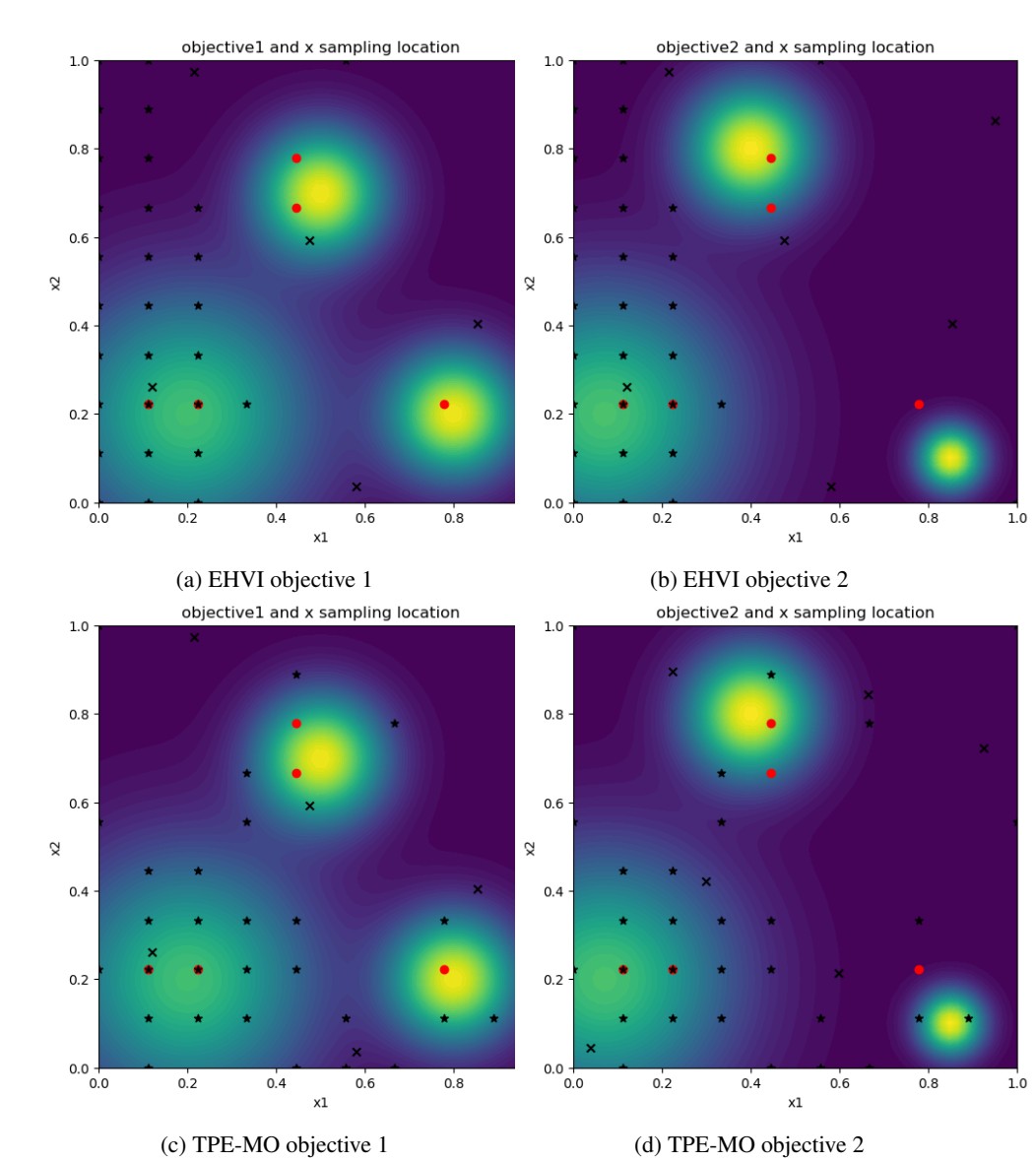

(a) EHVI objective 1

(b) EHVI objective 2

(c) TPE-MO objective 1

(d) TPE-MO objective 2

Figure 5: Searching behavior comparison in the design space for MOBO with EHVI and TPE-MO.

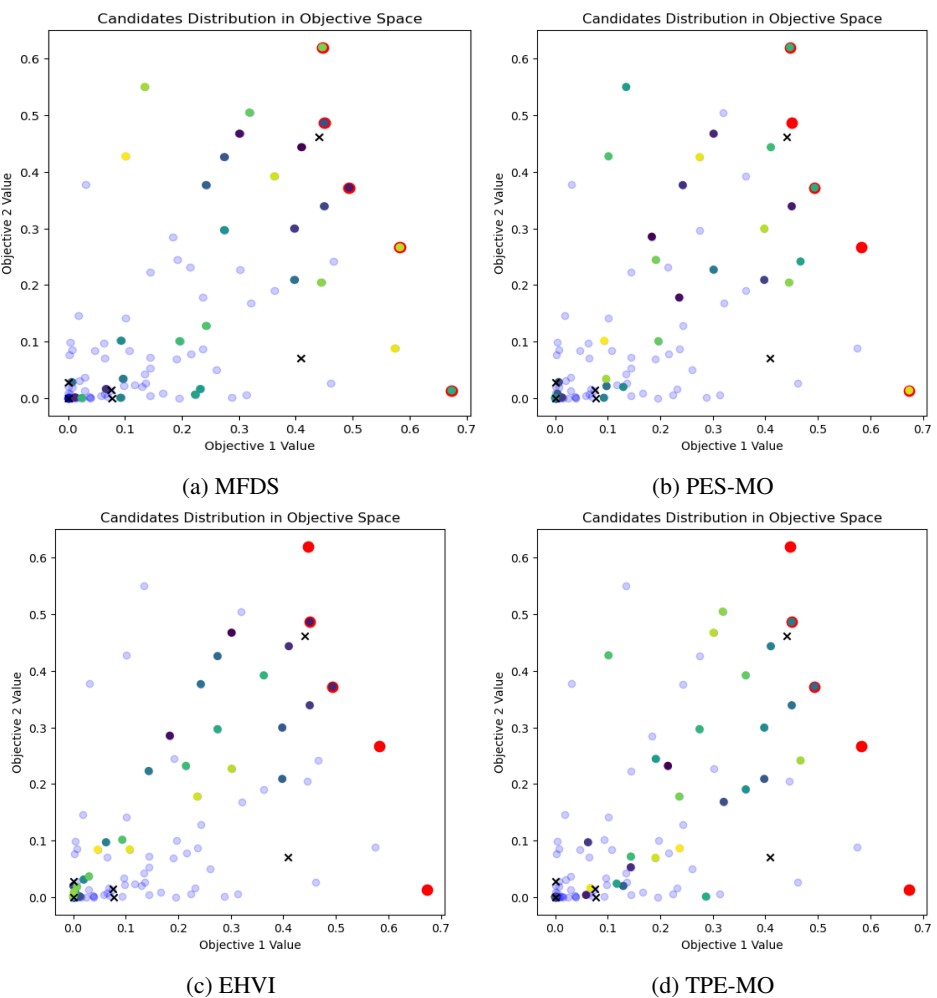

Figure 6: Searching behavior comparison in the objective space for GMM with different acquisition functions: (a) MFDS; (b) PES-MO; (c) EHVI; (d) TPE-MO. The Pareto front points are shown in red. Iteratively sampled points for each MOBO method are colored using the viridis colormap. Brighter yellow indicates points sampled closer to the final step. Darker shades represent points sampled earlier in the process. Points with transparent blue in the background indicate all the potential candidates. Black crosses are the initial data.

## A.3 ROBUSTNESS EXPERIMENTS ON DTLZ2

We conduct additional comparison experiments on DTLZ2 (Deb et al., 2002) under various settings. Fig. 7 shows the performance of three metrics mentioned in the main text. DTLZ2 is an objective function with a continuous and concave Pareto front. The true Pareto front lies on a unit hypersphere in the positive orthant of the objective space. Since DTLZ2 Pareto front consists of a single optimal region, exploitation-based methods (EHVI, EMMI, TPE-MO) outperform exploration-based methods (e.g. PES), as shown in Fig. 7. Meanwhile, MFDS remains competitive during the first 20 iterations, reinforcing our claim in the main text that MFDS fundamentally differs from PES and other exploration-driven approaches. Instead of focusing on optimal uncertainty reduction globally, MFDS prioritizes space-filling within the potential optimal region. Once a potential region is considered as fully explored (after 20 iterations), MFDS shifts its acquisition to identify the next promising region. In case DTLZ2, since there is no other optimal region, thus MFDS is outperformed by exploitation based method.

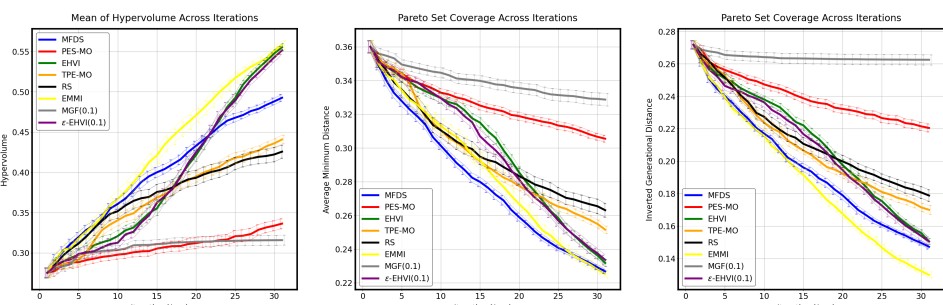

Figure 7: Performance comparison on DTLZ2 across different acquisition functions.

Fig. 8 illustrates the robustness of various acquisition functions across different noise levels (SNR = 2, 4, 8, 16). MFDS exhibits the highest noise tolerance, maintaining superior performance up to SNR = 2 before being outperformed by RS.

Besides MFDS, EMMI and TPE-MO also demonstrate relatively stable performance against noise at lower levels (SNR = 4, 8, 16). For EMMI, the acquisition follows a maximin criterion across each dimension, reducing the impact of noise compared to its effect on hypervolume. In TPE-MO, the expected improvement maxima is calculated based on the ratio of two density functions $l(x)$ and $g(x)$, where $l(x)$ represents the density estimated above a certain hypervolume threshold and $g(x)$ represents the density below that threshold (Ozaki et al., 2022). The noise only has a high influence on the samples that are close to the threshold. For samples that are far from the threshold, their probability of wrong classification is low, thus it won't have much effect on the density estimation and ratio function computation.

For methods that heavily rely on surrogate models (MGF, EHVI, and $\epsilon$-EHVI), performance degrades significantly under noise. In MGF, the exponential term amplifies noise, leading to instability. In EHVI and $\epsilon$-EHVI, noise affects both the surrogate model and the current hypervolume computation, making it difficult to obtain accurate improvement estimation.

MFDS remains stable against noise due to two key factors: (1) Similar to TPE-MO, estimation of the optimal probability term $p_{\min}$ can be considered as $l(x)$ estimation with a classification threshold using a rank 1 threshold. As long as the noise is not strong enough to alter the rank, the density estimation remains unchanged; (2) Uncertainty reduction relies on design space-filling, which is inherently robust to noise.

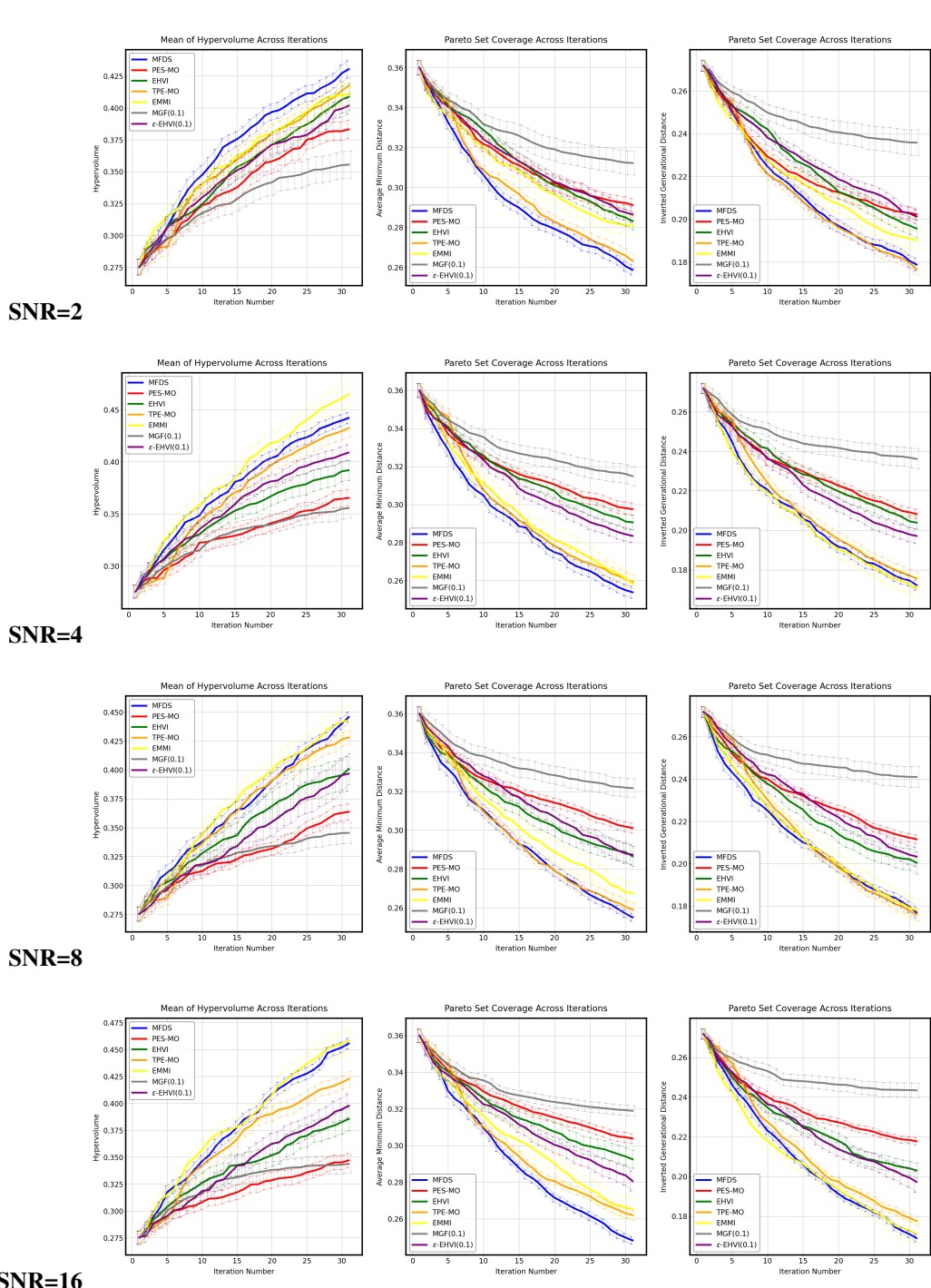

SNR=2

SNR=4

SNR=8

SNR=16

Figure 8: Performance comparison on DTLZ2 across different acquisition functions under varying noise levels.

## A.4   PROOF OF **LEMMA 4**

**Proof.**   We prove this by contraposition: suppose $\min_{\boldsymbol{x}'} G(\boldsymbol{x}' \cup \boldsymbol{X}_n, \mathcal{D}_n) > 0$, then $\exists \boldsymbol{x} \in \mathcal{X}$ s.t. $u_n(\boldsymbol{x}) > 0$. Denote $\boldsymbol{x}' = \arg\min_{\boldsymbol{x}} G(\boldsymbol{x} \cup \boldsymbol{X}_n, \mathcal{D}_n)$. In the following proof, we will prove $u_n(\boldsymbol{x}') > 0$ and by (17) , it is equivalent to prove:

$$\min_{\boldsymbol{x}'} G(\boldsymbol{x}' \cup \boldsymbol{X}_n, \mathcal{D}_n) > \mathbb{E}_{y|\boldsymbol{x}',\mathcal{D}_n}\Big[\min_{\boldsymbol{x}''} G(\{\boldsymbol{x}', \boldsymbol{x}''\} \cup \boldsymbol{X}_n, \mathcal{D}_n \cup \{\boldsymbol{x}', y\})\Big]. \tag{21}$$

Assume $G(\boldsymbol{x}' \cup \boldsymbol{X}_n, \mathcal{D}_n) > \delta > 0$. Since $G(\boldsymbol{x}' \cup \boldsymbol{X}_n, \mathcal{D}_n) = \mathbb{E}_{\boldsymbol{x}^*|\mathcal{D}_n}[d_{\min}(\boldsymbol{x}' \cup \boldsymbol{X}_n, \boldsymbol{x}^*)]$, the open set $A_\delta = \{\boldsymbol{x}|d_{\min}(\boldsymbol{x}' \cup \boldsymbol{X}_n, \boldsymbol{x}) > \delta\}$ must satisfy $p_{\min}(A_\delta|\mathcal{D}_n) > 0$. Denote the open ball with center $\boldsymbol{x}$ and radius $\delta$ as $B_\delta(\boldsymbol{x})$. The bounded search space $\mathcal{X}$ can be expressed as a union of $K < \infty$ open balls with radius $\delta$: $\mathcal{X} = \bigcup_{k=1}^K B_\delta(\boldsymbol{x}_k)$. With the distribution law, $A_\delta = \mathcal{X} \cap A_\delta = \bigcup_{k=1}^K (B_\delta(\boldsymbol{x}_k) \cap A_\delta)$. Since $p_{\min}(A_\delta|\mathcal{D}_n) > 0$, there $\exists k \leq K$ such that $p_{\min}(B_\delta(\boldsymbol{x}_k) \cap A_\delta|\mathcal{D}_n) > 0$. Denote $B_\delta(\boldsymbol{x}_k) \cap A_\delta$ as $A_\delta^{(k)}$. The right-hand side of (21) satisfies the following relation:

$$\mathbb{E}_{y|\boldsymbol{x}',\mathcal{D}_n}[\min_{\boldsymbol{x}''} G(\{\boldsymbol{x}', \boldsymbol{x}''\} \cup \boldsymbol{X}_n, \mathcal{D}_n \cup \{\boldsymbol{x}', y\})] \leq \mathbb{E}_{y|\boldsymbol{x}',\mathcal{D}_n}[G(\{\boldsymbol{x}', \boldsymbol{x}_k\} \cup \boldsymbol{X}_n, \mathcal{D}_n \cup \{\boldsymbol{x}', y\})]$$

$$= G(\{\boldsymbol{x}', \boldsymbol{x}_k\} \cup \boldsymbol{X}_n, \mathcal{D}_n) = \int_{\mathcal{X}} p_{\min}(\boldsymbol{x}^*|\mathcal{D}_n) d_{\min}(\boldsymbol{x}^*, \{\boldsymbol{x}', \boldsymbol{x}_k\} \cup \boldsymbol{X}_n) \mathrm{d}\boldsymbol{x}^* \tag{22}$$

In the last line of (22), the integral region $\mathcal{X}$ can be separate into two disjoint region: $A_\delta^{(k)}$ and $\mathcal{X}/A_\delta^{(k)}$. Since $d_{\min}(\boldsymbol{x}^*, \{\boldsymbol{x}', \boldsymbol{x}_k\} \cup \boldsymbol{X}_n) \leq d_{\min}(\boldsymbol{x}^*, \{\boldsymbol{x}'\} \cup \boldsymbol{X}_n)$, the integral in $\mathcal{X}/A_\delta^{(k)}$ can be bounded as:

$$\int_{\mathcal{X}/A_\delta^{(k)}} p_{\min}(\boldsymbol{x}^*|\mathcal{D}_n) d_{\min}(\boldsymbol{x}^*, \{\boldsymbol{x}', \boldsymbol{x}_k\} \cup \boldsymbol{X}_n) \mathrm{d}\boldsymbol{x}^*$$

$$\leq \int_{\mathcal{X}/A_\delta^{(k)}} p_{\min}(\boldsymbol{x}^*|\mathcal{D}_n) d_{\min}(\boldsymbol{x}^*, \{\boldsymbol{x}'\} \cup \boldsymbol{X}_n) \mathrm{d}\boldsymbol{x}^*, \tag{23}$$

Regarding the integral in $A_\delta^{(k)}$, we have $\forall \boldsymbol{x}^* \in A_\delta^{(k)}, d_{\min}(\boldsymbol{x}^*, \{\boldsymbol{x}', \boldsymbol{x}_k\} \cup \boldsymbol{X}_n) \leq |\boldsymbol{x}^* - \boldsymbol{x}_k| < \delta < d_{\min}(\boldsymbol{x}^*, \{\boldsymbol{x}'\} \cup \boldsymbol{X}_n)$. Hence:

$$\int_{A_\delta^{(k)}} p_{\min}(\boldsymbol{x}^*|\mathcal{D}_n) d_{\min}(\boldsymbol{x}^*, \{\boldsymbol{x}', \boldsymbol{x}_k\} \cup \boldsymbol{X}_n) \mathrm{d}\boldsymbol{x}^*$$

$$< \int_{A_\delta^{(k)}} p_{\min}(\boldsymbol{x}^*|\mathcal{D}_n) d_{\min}(\boldsymbol{x}^*, \{\boldsymbol{x}'\} \cup \boldsymbol{X}_n) \mathrm{d}\boldsymbol{x}^* \tag{24}$$

Combining (22)-(24), we can prove (21) as

$$\mathbb{E}_{y|\boldsymbol{x}',\mathcal{D}_n}[\min_{\boldsymbol{x}''} G(\{\boldsymbol{x}', \boldsymbol{x}''\} \cup \boldsymbol{X}_n, \mathcal{D}_n \cup \{\boldsymbol{x}', y\})]$$

$$< \int_{\mathcal{X}} p_{\min}(\boldsymbol{x}^*|\mathcal{D}_n) d_{\min}(\boldsymbol{x}^*, \{\boldsymbol{x}'\} \cup \boldsymbol{X}_n) \mathrm{d}\boldsymbol{x}^* = G(\{\boldsymbol{x}'\} \cup \boldsymbol{X}_n, \mathcal{D}_n). \qquad \square \tag{25}$$

That proves equation 21. $\square$

## A.5 ADDITIONAL EXPERIMENTAL DETAILS & DISCUSSIONS FOR RE3-5-4

For the RE3-5-4 problem, each candidate $\boldsymbol{x}$ is 5 dimensional, where $x_i \in [1,3]$ for each $i \in \{1, ..., 5\}$. These five variables specify the thickness of five reinforced components around the frontal structure of the vehicle. The first, second, and third objectives of the RE3-5-4 problem are to minimize the weight, acceleration characteristics, and toe-board instruction of the vehicle design (Tanabe & Ishibuchi, 2020). In our implementation, we negate all objective values, converting the problem into a multi-objective maximization task.

We adopt a different experimental setup compared to the one adopted for the GMM example. Given the need to explore a larger design space for an optimal practical solution, using a fixed set of 100 candidates, as in GMM, is unsuitable. Moreover, due to the $O(SN^3)$ computational complexity of MFDS, employing a large number of fixed candidates (e.g., 10k) would be inefficient. To address this, at each iteration, we sample 100 candidates from the design space using Latin Hypercube Sampling (LHS) (Helton & Davis, 2003). We then select the best candidate from the sampled subset, based on their acquisition function values. An essential step in computing the acquisition value is estimating the $G$ function. For each of the 100 candidates, we sample 100 times from the surrogate model and use the count of each candidate appearing on the Pareto front to estimate $p_{min}$. With estimated $p_{min}$, we can compute $G$ in both immediate and future rewards. The one-step-look-ahead sample size for future reward estimation is set to 5.

In Fig 9, we present the standard deviation (std) values of the hypervolume and AMD for the RE3-5-4 vehicle safety problem. Among the MOBO methods, MFDS demonstrates its robust performance with the lowest std values for both evaluation metrics. In contrast, PES-MO exhibits relatively higher std values for hypervolume, which can likely be attributed to its stronger exploratory and weaker exploitatory behavior. RS, EHVI, and TPE-MO show comparable hypervolume std performances. For the AMD, the std values by MFDS, PES-MO, and RS follow a similar decreasing trend. However, EHVI and TPE-MO, both based on the EI-based acquisition functions, have higher variance when the number of iterations increases, suggesting that the Pareto set convergence is difficult to achieve as we have discussed in the main text.

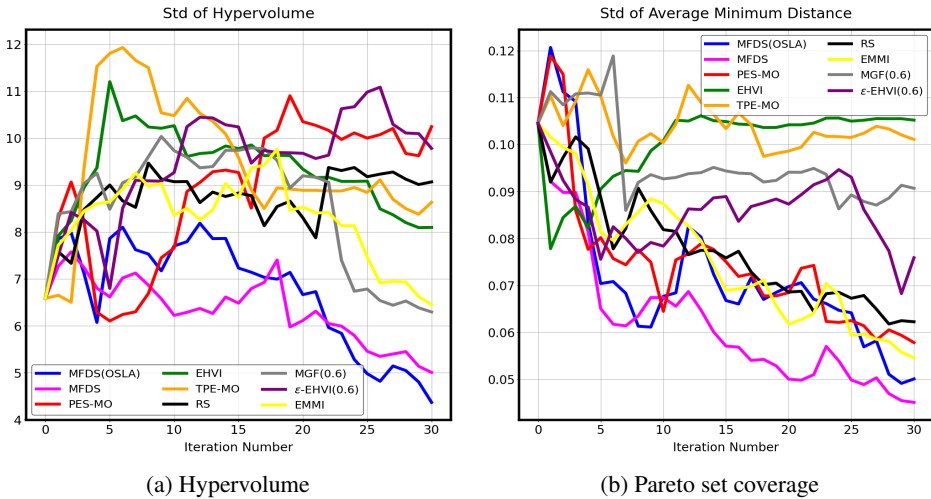

| (a) Hypervolume | (b) Pareto set coverage |

Figure 9: Observed std values of different MOBO methods for the case study RE3-5-4: (a) Hypervolume; (b) Average minimum distance.

Fig. 10 illustrates the search patterns of various MOBO methods with the corresponding acquisition functions for the RE3-5-4 problem. Our MFDS method again demonstrates comprehensive coverage on most of the Pareto front regions. PES-MO also explores most of the Pareto front, but shows limited exploitation in these regions compared to MFDS. EHVI tends to get stuck in a local Pareto region near the top, while TPE-MO improves coverage over EHVI but still lacks exploration in the lower part of the Pareto front. None of the methods fully capture the lower Pareto region, likely due to the scarcity of candidates in this area.

Given the stochastic nature of the candidate selection process, achieving a perfect match between the true Pareto front (shown in red in Fig. 10) and the approximate Pareto front identified by MOBO with the corresponding acquisition functions (depicted as black points with yellow stars in Fig. 10) is challenging. Instead of directly counting how many points from the Pareto front are covered by the suggestions, we evaluate the coverage by counting how many Pareto points lie within a hypersphere centered at the best suggestions, with a radius of $r = 0.12$.

The numbers of covered Pareto front points are 9, 8, 2, 1, and 0 for MFDS, TPE-MO, EHVI, PES-MO, and RS, respectively. Notably, while PES-MO appears to provide the decent coverage in Fig. 10, the actual suggestions are relatively far from the true Pareto front, indicating its poor exploitation capability as explained in the main text. In contrast, TPE-MO exhibits good overlap with the Pareto front, though this is primarily concentrated in the central region of the objective space. EHVI demonstrates limited coverage due to its tendency to focus on regions where there are few Pareto-optimal candidates, particularly getting stuck near the top corner of the objective space.

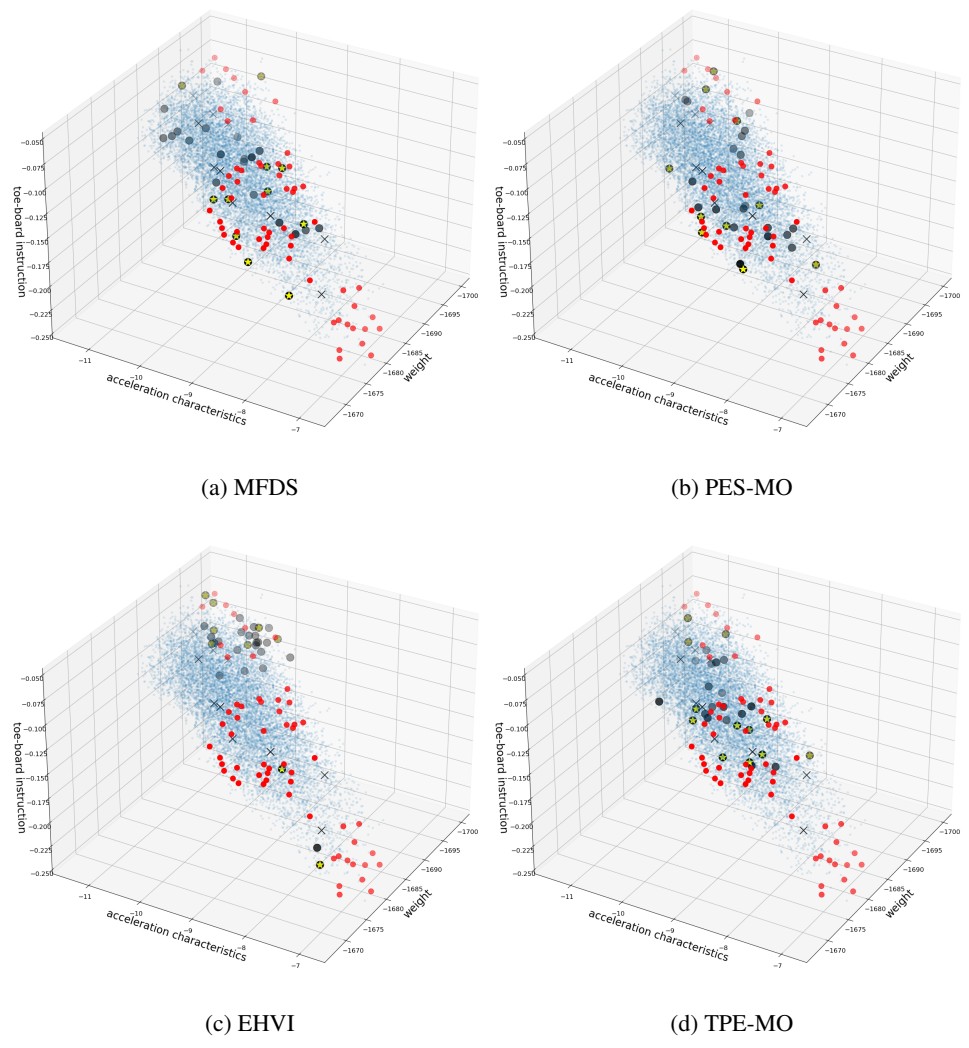

Figure 10: Searching behavior comparison in the objective space for RE3.5.4 with different acquisition functions: (a) MFDS; (b) PES-MO; (c) EHVI; (d) TPE-MO. The Pareto front points are shown in red. Iteratively sampled points are marked as black dots for different MOBO methods. Points with transparent blue in the background indicate all the potential candidates. Black crosses are the initial data. Yellow stars on black dots represent the approximate Pareto front, highlighting the best trade-off solutions identified by each acquisition function.

## A.6 Experimental settings, run-time, & source code

We compare MFDS against EHVI (Daulton et al., 2020; 2021), PES-MO (Garrido-Merchán & Hernández-Lobato, 2019; Hernández-Lobato et al., 2014), Multiobjective Tree-structured Parzen Estimator (TPE-MO) (Ozaki et al., 2020; 2022), $\epsilon$-EHVI (Bartz-Beielstein & Zaefferer, 2017), Moment-Generating Function (MGF) (Wang et al., 2017), EMMI (Olofsson et al., 2018) and random sampling (RS). RS serves as a baseline for optimization performance. TPE-MO, an EI-based acquisition function, offers a lower computational cost compared to EHVI. All methods, except TPE-MO, are implemented within the BoTorch framework (Balandat et al., 2020), while TPE-MO is implemented using Optuna (Akiba et al., 2019).

For the GMM problem, all the computations were performed on an Intel(R) Xeon(R) Gold 6248R CPU. This system features 24 cores per socket, with a total of 48 cores. The average time per iteration for the MFDS method was 32.3 seconds.

For the RE3-5-4 problem, the trials were distributed between two systems. Half of the trials (10 out of 20) were executed on the Intel(R) Xeon(R) Gold 6248R CPU, with an average iteration time of 65.2 seconds for MFDS. The remaining 10 trials were conducted on an Intel(R) Xeon(R) Platinum 8352Y CPU, which operates at 2.20GHz and has 32 cores per socket, with a total of 64 cores. The average iteration time on this system was 59.1 seconds for MFDS. The workload was distributed across these two systems to reduce the overall time cost of the experiments.

Table 1: Run-time for MOBO with different acquisition functions

| Run-time per iteration (seconds) | | |
|---|---|---|
| Acquisition function | GMM | RE3-5-4 |
| RS | 0.06 | 0.07 |
| TPE-MO | 0.2 | 0.83 |
| EMMI | 0.85 | 1.06 |
| EHVI | 0.95 | 1.34 |
| MGF | 1.32 | 1.91 |
| PES-MO | 15.4 | 26.3 |
| MFDS(OSLA) | 32.3 | 65.2 |

Table 1 illustrates the computation times required for each acquisition function per iteration. RS was the quickest, as anticipated. TPE-MO demonstrated relatively short computation time compared to EHVI, primarily due to the faster kernel density estimation compared to GP fitting. PES-MO and MFDS incurred significantly higher computation time, being one-step-look-ahead methods. The time scale is influenced by the number of samples used for one-steplook-ahead predictions. In PES-MO, researchers typically employ expectation propagation (EP) (Hennig & Schuler, 2012) to derive an analytical form for estimating the acquisition function. Despite EP's $O(N^4)$ complexity, it remains faster than the $O(SN^3)$ Monte Carlo calculations for all candidates. Thus, PES-MO with EP has a lower time cost compared to MFDS with Monte Carlo estimation.

## A.7 Use of Large Language Models

Large Language Models are only used to check vocabulary and grammar for polishing purpose.

