# OpenReview forum: "Bayesian Optimization by Minimum Filling Distance Search"
_ICLR.cc/2026/Conference — Submitted to ICLR 2026_

### Official Review · Reviewer_XTdq · 2025-10-27

**Soundness:** 3
**Presentation:** 3
**Contribution:** 2
**Rating:** 4
**Confidence:** 4

**Summary:**

This paper proposes a new acquisition function for Single-Objective (SO) and Multi-Objective (MO) Bayesian Optimization (BO), designed to minimize the expected closest distance between an optimal input and a corresponding reference set. In both the SO and MO settings, the acquisition function is formulated as a one-step look-ahead approach. The proposed method has been evaluated on multiple synthetic benchmark functions, demonstrating its effectiveness in improving solution quality and Pareto front coverage.

**Strengths:**

- The proposal of leveraging the average minimum distance as a metric for formulating the acquisition function is, to the best of my knowledge, novel and interesting.
- The paper considers both the SO and MO settings and demonstrates that a similar formulation can be applied, with experiments cover both settings.

**Weaknesses:**

- Limited synthetic benchmarks: The paper benchmarks on both single-objective functions (Gaussian mixture, Shekel, Forrester) and multi-objective problems (2-dimensional GMM, DTLZ2, and one real-world problem RE3-5-4) which is plausible. However, the benchmark complexity is more of proof of concept level, and the claim could be much more strength with more complicated empirical comparison.
- Unclear attribution of performance gain: One unclear aspect is that the authors directly start with a one-step look-ahead formulation (Eq. 14, Eq. 17). Since one-step look-ahead methods already provide a performance benefit, it is not clear whether the performance gain of this algorithm stems from the new acquisition function formulation or from the look-ahead strategy itself. To ensure fair comparison, the standard acquisition functions should also be evaluated with one-step look-ahead. If the authors believe the look-ahead strategy is a core contribution, an ablation study should be included.
- The complexity and scalability: the algorithm has its fundamental complexity and scalability issue, which would restrict its apllicability in moderate-high input dimensionalities.

**Questions:**

- How is the  $X_n$ is set if it is different from $X$ ?

---

> ### Author Response · Authors · 2025-11-21
>
> ## Detailed Responses to Reviewer XTdq
>
> We truly appreciate your constructive critiques and suggestions. In the following, we provide our point-by-point responses to address the raised concerns.
>
> ### **Weaknesses:**
>
> **the claim could be much more strength with more complicated empirical comparison.**
>
> We incorporated additional experimental results, including two real-world MOBO benchmark problems: RE2-4-1, which involves 4 input variables and 2 output objectives [1], and RE3-7-5, which includes 7 inputs and 3 outputs [2]. For RE3-7-5 problem, besides 3 objectives, there are also 11 constraints which make the searching task more complicated. The results show that MFDS remains effective in higher-dimensional settings and consistently outperforms other acquisition strategies. We provided more details of these two problems and results discussion in Section 5.3 in the main text revision. Meanwhile, we are extending our experiments to more complex settings, including higher dimensional inputs and additional objectives. Some baselines become significantly more time consuming as the number of objectives increases, for example, EHVI is well known to suffer from computational complexity issues in high dimensional objective spaces [3]. If we are able to finish running all nine baselines in time, we will report the results.
>
> **An ablation study should be included.**
>
> In our main results, we have compared MFDS with its one-step-look-ahead (OSLA) variant. The experiments consistently show that the performance gains come mainly from MFDS itself for both single objective and multi-objective cases, while the additional OSLA component can provide marginal improvement. We have emphasized the difference between MFDS and MFDS(OSLA) in Section 5 of the revised main text.
>
> **The complexity and scalability: the algorithm has its fundamental complexity and scalability issue.**
>
> While the computational complexity is indeed high, we are confident that it can be significantly reduced through analytical approximations. As shown in Table 1 of Appendix A.6, the theoretical complexity is comparable to PES. Over the years, researchers have developed efficient analytical approximations that make PES faster. We are actively exploring strategies to derive a closed-form or near closed-form approximation for MFDS. More importantly, in real-world applications, the main concerns are often that the actual experimental cost and time can be much higher compared to computational overheads by MFDS or other MOBO methods. In the seven dimensional, three objective setting, the computational cost is approximately 60 seconds. Since real-world evaluations or simulations typically require far more than 60 seconds, this computational overhead is not a significant concern.
>
>
>
> ### **Questions:**
>
> **How is the $X_n$ is set if it is different from $X$?**
>
> Typically, $X_n$ is the same as $X$. We emphasized the distinction only to clarify that the estimation of $p\_{min}$ is based on a surrogate model fitted to a dataset $\mathcal{D} = \\{X, Y\\}$, which may not always match the set of locations already selected by the acquisition function. For example, if a pretrained surrogate is used, $X$ may differ from $X_n$. However, in the standard setting where the surrogate is trained solely on the current dataset $\mathcal{D}_n = \\{X_n, Y_n\\}$, we simply have $X = X_n$.
>
> [1] https://scholarsmine.mst.edu/civarc_enveng_facwork/2955/
>
> [2] https://link.springer.com/article/10.1007/s00158-002-0247-6
>
> [3] https://arxiv.org/abs/2306.00344

---

### Official Review · Reviewer_p3DZ · 2025-10-31

**Soundness:** 3
**Presentation:** 3
**Contribution:** 3
**Rating:** 6
**Confidence:** 3

**Summary:**

The authors propose to use as an acquisition function for Bayesian optimization the expected minimum distance between the design points and the true minimizer.
They develop a convergence proof and demonstrate this method on both single and multi-objective problems.

**Strengths:**

It's nice to have some theoretical results.
Namely, the authors show that the acquisition function converges to zero.
It would be more ideal to have a bound/rate on regret or something like this but since there are also fairly extensive numerical results I think this is enough.

The numerical results seem adequate, with a decent number of benchmark comparators, uncertainty quantification on the iteration number, and five benchmark applications, which is perhaps sufficient.

**Weaknesses:**

I found some of the discussion about exisiing BO methods surprising, and I think readers would appreciate more context for these claims:

i) "As a result, when an inappropriate prior is set for the probabilistic model, BO may fall in local optima (Wang & de Freitas, 2014)." What part of that article are you specifically referring to when you say that inappropriate priors lead to local optima?

ii) "[Entropy Search] design works well for objective functions with a single global optimum. For objective functions in real-world applications, however, there is often not just one single optimum." I think I unerstand why, conceptually, ES assumes a single global optimum. But have we actually observed any failure cases empirically in the multi-optimum case? Since it is well defined even if one of the local optima is only epsilon worse than the other, it's not obvious to me that it would actually fail in practice. Furthermore, the subsquent Max-Value entropy search of Wang and Jegelka would seemingly solve this anyways unless I'm issing something.

I like some of the intuition built in the intro about why this method is useful under model misspecification, but I ultimately did not fully understand the derivation of the acquisition function. It would be helpful to in particular say a bit more about how the second term in the definition was obtained.

Though a reasonable idea conceptually, there does not appear to be much in the way of analyitcal tractability for this acquisition function, and the authors' experiments are on small iteration counts. This will limit the applicability of the method.

**Questions:**

1) I'm a bit confused with the main definition of the acquisition function u at the bottom of page 4.  The first term does not seem to depend on x. Is it supposed to be D_n\cup\{x,y\} instead of just D_n?

2) At the end of the day, X_N is going to be the set of previously observed design points at time N, or not?

3) Doesn't this method suffer from a similar assumption of a unique global optimum as you mention in your criticism of ES?

---

> ### Author Response · Authors · 2025-11-21
>
> ## Detailed Responses to Reviewer p3DZ
>
> We truly appreciate your constructive critiques and suggestions. In the following, we provide our point-by-point responses to address the raised concerns.
>
> ### **Weaknesses:**
>
> **I found some of the discussion about exisiing BO methods surprising, and I think readers would appreciate more context for these claims.**
>
> We are glad that our discussion of prior work caught your attention. We originally included a detailed section analyzing the pros and cons of almost all major existing BO acquisition strategies, but due to the page limit we moved it to Appendix A.1. We hope this more detailed literature review is useful for readers. Also, since we have examined more than ten baseline methods, we tried our best to clarify the strengths and weaknesses of each approach, and we welcome further discussions and suggestions how we may further improve Appendix A.1.
>
> **What part of the Wang & de Freitas (2014) shows that inappropriate priors lead to local optima?**
>
> In Wang & de Freitas (2014), Figure 1 shows that a mis-specified GP prior causes BO with the EI acquisition function (left column) to be trapped in a suboptimal region, failing to converge to the true maximum. The toy Gaussian mixture example we used in our demo section is a case with mis-specified GP prior. The GP with the SE kernel is taken as the surrogate model, with the hyperparameters set to $\sigma_l = 0.01$, $\sigma_f = 1$ throughout the BO procedure. Small $\sigma_l$ here simulates the situation in which an observation point can only provide objective value information for points nearby. As a result BO with some acquisition function may only sample points close to each other. We have added this description in Section 5.1 of the revised main text.
>
> **Have we actually observed any failure cases empirically in the multi-optimum case for ES methods?**
>
> Beyond the examples shown in our main text, there are multiple studies that demonstrated empirical failure cases of ES. Several studies have shown that ES can exhibit a very slow convergence rate [1,2]; and in some challenging real-world problems, e.g. the Penicillin benchmark, its convergence can be even slower than random sampling as reported in [3]. The root cause is fundamental: ES cannot distinguish between regions that are definitely optimal and regions that are definitely impossible, because both yield nearly zero information contribution ($p\_{min}\log p\_{min}$). Once ES prematurely assigns a region as “certain,” it stops sampling there and fails to verify its true objective value, leading directly to these empirically observed failures.
>
> **It would be helpful to in particular say a bit more about how the second term in the definition was obtained.**
>
>
> Generally, the sequential decision problem can be expressed and solved with a dynamic programming formulation, which inspires our acquisition function. The state is the observation set $\mathcal{D}\_n$, and the policy $\pi\in\Pi$ is defined as $\boldsymbol{x}\_{n+1} = \pi(\mathcal{D_n})$. Define the value function at iteration $n$ as:
>
> $$
> V\_n(\mathcal{D}\_n) = \inf_\{\pi\in\Pi} \mathbb{E}^{\pi}[G(\boldsymbol{X}\_N, \mathcal{D}\_N) | \mathcal{D}\_n]
> $$
>
> When \(n = N\), the value function is simply
>
> $$
> V_N(\mathcal{D}_N) = G(\boldsymbol{X}_N, \mathcal{D}_N).
> $$
>
> Based on the Bellman equation [4], the value function for $0 \le n < N$ can be computed recursively by
>
> $$
> V\_n(\mathcal{D}\_n)
> = \min\_{\boldsymbol{x}}  \mathbb{E}_{y |  \boldsymbol{x}, \mathcal{D}\_n}
> \left[ V\_{n+1}(\mathcal{D}\_n \cup \{(\boldsymbol{x}, y)\}) \right].
> $$
>
> For $n = N-1$, the optimal policy reduces to a greedy rule:
>
> $$
> V\_{N-1}(\mathcal{D}\_{N-1})
> = \min\_{\boldsymbol{x}}  \mathbb{E}_{y | \boldsymbol{x}, \mathcal{D}\_{N-1}}
> \left[ G(\boldsymbol{x} \cup \boldsymbol{X}\_{N-1},  \mathcal{D}\_{N-1} \cup \{(\boldsymbol{x}, y)\}) \right]
> = \min\_{\boldsymbol{x}} G(\boldsymbol{x} \cup \boldsymbol{X}\_{N-1}, \mathcal{D}\_{N-1}).
> $$
>
> For $n = N-2$, the value function corresponds to the one-step-look-ahead policy:
>
> $$
> V\_{N-2}(\mathcal{D}\_{N-2})
> = \min\_{\boldsymbol{x}} \, \mathbb{E}_{y | \boldsymbol{x}, \mathcal{D}\_{N-2}}
> \left[ V\_{N-1}(\mathcal{D}\_{N-2} \cup \{(\boldsymbol{x}, y)\}) \right]
> $$
>
> $$
> = \min\_{\boldsymbol{x}} \, \mathbb{E}_{y \,|\, \boldsymbol{x}, \mathcal{D}\_{N-2}}
> \left[ \min\_{\boldsymbol{x}''} G(\{\boldsymbol{x}, \boldsymbol{x}''\} \cup \boldsymbol{X}\_{N-2}, \, \mathcal{D}\_{N-2} \cup \{(\boldsymbol{x}, y)\}) \right].
> $$
>
> We have included this explanation in Section 3 of the revised main text, under the Single-objective MFDS paragraph to make our formulation development context more integrated as reviewer Lh5F also has similar questions about how our acquisition strategy is developed.

---

> ### Author Response · Authors · 2025-11-21
>
> **There does not appear to be much in the way of analytical tractability for this acquisition function.**
>
> Indeed, deriving closed-form acquisition function is challenging when considering the combination of $p\_{min}$ and filling space strategies. However, prior work shows that the analytical form of $p\_{min}$ can be approximated using expectation propagation [5], and that space-filling strategies can be modeled as linear functions within regions partitioned by existing observation points. Therefore, combining the analytical form of $p\_{min}$ with a space-filling approach is feasible.
>
>
> ### **Questions:**
>
> **I'm a bit confused with the main definition of the acquisition function u at the bottom of page 4. The first term does not seem to depend on x. Is it supposed to be $D_n\cup{(x,y)}$ instead of just $D_n$?**
>
> Yes, your statement is correct: the first term does not depend on $x$. Our acquisition function follows a structure similar to PES, where the first term serves as a state function that represents the current state of knowledge, and the second term varies with $x$ to quantify the potential state improvement gained by evaluating that location. In PES, this state function is the entropy of the optimizer. In our case, the state function is the expected minimum distance to the global optimizer.
>
> **At the end of the day, X_N is going to be the set of previously observed design points at time N, or not?**
>
> Yes, $X\_N$ is simply the set of design points observed up to iteration $N$. We assume your question arises from the distinction between $X_n$ and $X$. This distinction is clarified in detail in our response to reviewer XTdq.
>
> **Doesn't this method suffer from a similar assumption of a unique global optimum as you mention in your criticism of ES?**
>
> As discussed earlier, the core limitation of ES is its inability to distinguish between deterministic regions: both definitely optimal and definitely non-optimal locations contribute zero entropy. In contrast, our acquisition directly estimates $p\_{min}$ directly rather than its entropy. Consequently, in the scenarios with multiple global optima, all regions with high probability of being optimal naturally receive high $p\_{min}$. We then use the proximity term to decide which region to acquire first. Once a region is explored, its proximity term decreases, allowing the algorithm to continue exploring the remaining high probability regions. Therefore, multiple global optima do not pose a problem for MFDS.
>
> [1] https://arxiv.org/pdf/2006.05078
>
> [2] https://proceedings.mlr.press/v119/zhang20i/zhang20i.pdf
>
> [3] https://arxiv.org/pdf/2306.00344
>
> [4] https://web.mit.edu/dimitrib/www/dpchapter.pdf
>
> [5] https://arxiv.org/abs/2206.04771

---

### Official Review · Reviewer_Lh5F · 2025-11-01

**Soundness:** 3
**Presentation:** 3
**Contribution:** 2
**Rating:** 4
**Confidence:** 3

**Summary:**

The paper proposes a new acquisition function for Bayesian Optimization, Minimum Filling Distance Search (MFDS). MFDS chooses the next point by minimizing the expected distance between the sequence of sampled locations and (i) the posterior distribution of the global minimizer (single‑objective) or (ii) the posterior distribution of the Pareto set (multi‑objective). This explicitly leverages where past samples are, aiming to avoid oversampling and to cover regions with high probability of optimality. The authors prove an asymptotic convergence result for the single‑objective case (under specific assumptions) and empirically compare MFDS to other popular acquisition functions.

**Strengths:**

- Nice and intuitive geometric representation of the objective. This acquisition function makes sense.
- Theoretical proof of the convergence.
- Results showing that this method outperforms others (though more tests should be performed).

**Weaknesses:**

- Theory is limited just to 1D.
- Claiming theoretical proof of convergence and then mentioning the "almost surely" converges should be clarified better.
- Very limited tests are performed. The authors should use more test functions and average results across different seeds.

**Questions:**

- Fig.1 why don't you show comparison between different acquisition functions of the next selection point where the set of all the current points is the same? Current comparison seems to be unfair.
- why are n-1 iterations done in one way and the last iteration is a greedy approach? This seems like a heuristic.

---

> ### Author Response · Authors · 2025-11-21
>
> ## Detailed Responses to Reviewer Lh5F
>
> We truly appreciate your constructive critiques and suggestions. In the following, we provide our point-by-point responses to address the raised concerns.
>
> ### **Weaknesses:**
>
> **Theory is limited just to 1D.**
>
>
> Our theoretical proof focuses on the single-objective case. Extending convergence proofs to multi-objective BO is known to be much more difficult, and only a very small number of works provide any theoretical guarantee of existing MOBO strategies. Among the few existing references, the most commonly cited results are based on EHVI-type methods. However, similar to the classical EI convergence proof, these guarantees rely on strong assumptions, particularly the condition of maximum information gain, which essentially requires the entire domain to be sufficiently explored [1,2,3]. This assumption is significantly stronger than those used in our theoretical result, and it limits the practical applicability of such analyses.
>
> We will also explore whether our proof techniques can be extended to multi-objective BO, with the goal of contributing further theoretical advances to the BO community in future work.
>
> **Claiming theoretical proof of convergence and then mentioning the "almost surely" converges should be clarified better.**
>
>
> Our result provides an almost sure (a.s.) convergence guarantee, which is the standard form of theoretical convergence in Bayesian optimization under stochastic GP models. Since acquisition decisions depend on random GP sample paths, deterministic convergence is generally not attainable. Theorem 1 emphasizes that the proposed algorithm will not get stuck in a suboptimal local optimum except on a probabity-zero event. We clarify that our use of “theoretical convergence” specifically refers to this a.s. convergence and revise the text accordingly.
>
> **Very limited tests are performed. The authors should use more test functions and average results across different seeds.**
>
> We incorporated additional experimental results, including two real-world MOBO benchmark problems: RE2-4-1, which involves 4 input variables and 2 output objectives [4], and RE3-7-5, which includes 7 inputs and 3 outputs [5]. The results show that MFDS remains effective in higher-dimensional settings and consistently outperforms other acquisition strategies. We provided more details of these two problems and results discussion in Section 5.3 in the main text revision.
>
> For all methods in our experiments, we perform 20 independent trials using different random seeds. The main results report the mean (solid line) and standard deviation (dotted line) across these trials. Therefore, the strong performance of MFDS shown in the main text should be statistically solid and reliable.

---

> > ### Author Response · Authors · 2025-11-21
> >
> > ### **Questions:**
> >
> > **Fig.1 why don't you show comparison between different acquisition functions of the next selection point where the set of all the current points is the same? Current comparison seems to be unfair.**
> >
> > We agree that comparing different acquisition functions using the same set of current points would appear ideal. In fact, we already ensure fairness by starting all methods from the same initial design (red crosses) in Fig. 1, so all approaches begin from identical information.
> >
> > However, forcing all methods to share the exact same intermediate “current set” for only the next selection is not suitable for illustrating the specific failure modes we analyze. BO acquisitions are inherently sequential, and their characteristic behaviors (e.g., over-exploitation or insufficient exploration) only become apparent over multiple steps. As a result, different methods require different scenarios to demonstrate their limitations.
> >
> > For PES, its issue is that it may fail to acquire near the global optima even when that region has already been explored. Therefore, the demonstration requires a scenario where the global optimum region is sufficiently sampled beforehand.
> >
> > For EI, its limitation is over-sampling, which can prevent it from exploring the global optimum region at all. To reveal this behavior, we need a scenario where the global optimum region is not yet explored.
> >
> > These two scenarios inherently contradict with each other and cannot be shown using the same single “current” setup. For this reason, instead of constructing artificial and inconsistent intermediate states, we use the same initial design for all methods and allow each acquisition function to run sequentially for multiple iterations.
> >
> > **why are n-1 iterations done in one way and the last iteration is a greedy approach? This seems like a heuristic.**
> >
> > In our dynamic-programming formulation, the final step $n=N-1$ is special because no future decisions remain. As a result, the value function at this step reduces directly to evaluating the objective itself, without having to consider any future expectation. Therefore, the optimal policy at the last step becomes a simple greedy choice. In contrast, all earlier steps $n<N-1$ must account for how the current decision influences future decisions. Their value functions include expectations over the next observation, leading to look-ahead acquisition rules rather than a greedy one. This is why the acquisition function at the last step appears different: only at that step does the dynamic-programming recursion terminate, eliminating future uncertainty. Further details on the development of acquisition strategies are provided in Section 3 of the revised main text, under the Single-objective MFDS paragraph, as well as in our response to reviewer p3DZ.
> >
> > [1] https://arxiv.org/abs/2411.03641
> >
> > [2] https://proceedings.mlr.press/v119/zhang20i.html
> >
> > [3] https://proceedings.mlr.press/v180/daulton22a.html
> >
> > [4] https://scholarsmine.mst.edu/civarc_enveng_facwork/2955/
> >
> > [5] https://link.springer.com/article/10.1007/s00158-002-0247-6

---

### Official Review · Reviewer_9e8V · 2025-11-02

**Soundness:** 3
**Presentation:** 3
**Contribution:** 3
**Rating:** 6
**Confidence:** 1

**Summary:**

This paper proposes an acquisition function for Bayesian optimization. The proposed acquisition function considers the minimum distance between the sampling path and the set of optimal solutions. This acquisition function can also be used for multi-objective optimization. The authors prove the convergence of the proposed method and demonstrate its superior performance through experiments.

**Strengths:**

- The problem is well-motivated, and considering the past observations is intuitively important for Bayesian optimization.
- Experimental results demonstrate better performance of the proposed method than existing baseline methods.

**Weaknesses:**

I am not an expert in Bayesian optimization, so I could not fully understand the advantage of the theoretical result over existing theoretical guarantees in this field. In particular, I did not fully understand what benefit we obtain from Theorem 1. Does $\lim_{n\to \infty} \min_{x'} G(x' \cup X_n, \mathcal{D}_n) = 0$ indicate that $x_n$ converges to the optimal solution(s)? Is there a similar convergence guarantee for other acquisition functions?

**Questions:**

My main concerns and questions are outlined in the Weaknesses section. Additionally, I have the following questions:

- In Eq. (8), what is the definition of $P$?
- In Eq. (12), it seems difficult to compute the integral involving $\mu$ exactly. How can we compute this integral? What is the computational complexity?
- In Section 5.1, what does the equation $f_{GM}(x) = -0.5,\mathcal{N}(-\mu, 2\sigma^2) - 0.55,\mathcal{N}(\mu, \sigma^2) + 1$ mean? Does it mean that the probability density functions of the Gaussian distributions $\mathcal{N}(-\mu, 2\sigma^2)$ and $\mathcal{N}(\mu, \sigma^2)$ are used?

---

> ### Author Response · Authors · 2025-11-21
>
> ## Detailed Responses to Reviewer 9e8V
>
> We truly appreciate your constructive critiques and suggestions. In the following, we provide our point-by-point responses to address the raised concerns.
>
> ### **Weaknesses:**
>
> **I did not fully understand what benefit we obtain from Theorem 1. Does $\lim_{n\to \infty}\min_{x'}G(x'\cup X_n, \mathcal{D_n})$
>  indicate that $x_{n}$ converges to the optimal solution(s)? Is there a similar convergence guarantee for other acquisition functions?**
>
>
> Thank you for carefully examining our proof. This is a great question. In our setting, $G$ function converging to 0 implies that the evaluated design points $X_n = \\{x\_1, ..., x\_n\\}$ get arbitrarily close to the global optimum $x^*$ (or more generally, to the set of global optima). Thus Theorem 1 provides a standard asymptotic global-convergence gurantee which indicates that the algorithm does not get stuck at a local optimum.
>
> We agree that similar types of convergence guarantees exist for other acquisition functions, such as EI and UCB, under appropriate assumptions. Therefore, we do not claim that Theorem 1 alone makes our algorithm strictly stronger than these existing algorithms. Rather, the main advantage of our method lies in its robustness-oriented design which incorporates a space-filling component, and in its strong empirical performance. The theoretical result confirms that these design choices preserve global convergence and avoids local-optimum trapping that purely greedy heuristics may suffer.
>
> ### **Questions:**
>
> **In Eq. (8), what is the definition of $P$?**
>
> We use $P$ to represent the probability of observing $y$ conditioned on $\boldsymbol{x}$ and $D$. Since it is independent of $x^*$, we can take it out and its integral is $1$.
>
> **In Eq. (12), it seems difficult to compute the integral involving $\mu$ exactly. How can we compute this integral? What is the computational complexity?**
>
> We do not compute the integral involving $\mu$ directly. As shown in many previous works, integrating $p_{min}$ over any $\mu$ is intractable and does not admit an analytical form. Therefore, we use MC sampling to estimate $p_{min}$ and the integral as previously done. The detailed complexity analysis can be found in Appendix A.6. Generally if the number of observation locations is $N$ and the number of MC samples is $M$, the total computational complexity is $O(N^2 M)$, where the $N^2$ term comes from identifying the Pareto optimal set. The computation cost of the distance term is negligible compared with the cost of estimating $p_{min}$. This $O(N^2 M)$ complexity is the same to that of PES, and should likewise be practical for many real-world BO applications.
>
> **In Section 5.1, what is the exact definition of GM toy example?**
>
> We use additive mixtures of Gaussian (exponential quadratic) functions in the toy example in Section 5.1. We created such Gaussian mixtures so that the blackbox BO task is to find the two valleys represented by the corresponding components, simulating a simple scenario that requires both exploration and exploitation. We provided more details of this toy example in Section 5.1 in the main text revision.

---

### Author Response · Authors · 2025-12-03
**Rebuttal Summary**

Dear Program Committee, Area Chairs, and Reviewers,

We are thankful for the time and constructive feedback provided by the reviewers. To best support the newly assigned AC and facilitate their evaluation, we have summarized the key points from the reviewers and the subsequent reviewer-author discussions below.

### **Strengths:**

The table below summarizes the core strengths identified by the reviewers.

| Category | Specific Strengths Mentioned | Relevant Reviews |
| :--- | :--- | :--- |
| **Empirical Performance** | The method demonstrates **superior performance** in experimental/numerical results, **outperforming existing baseline methods**. Results are **adequate/extensive** for validation. | 9e8V, Lh5F, p3DZ |
| **Novelty & Intuition** | The proposal of leveraging the **average minimum distance** for the acquisition function is **novel and interesting**. The function is described as **nice, intuitive, and makes sense**. The problem is **well-motivated**, as considering past observations is **intuitively important**. | 9e8V, Lh5F, XTdq |
| **Theoretical Rigor** | **Theoretical proof of convergence** is provided, specifically showing the acquisition function **converges to zero**. | Lh5F, p3DZ |
| **Scope & Generality** | The paper considers **both Single-Objective (SO) and Multi-Objective (MO) settings**, demonstrating that a similar formulation can be broadly applied across optimization regimes. | XTdq |

Most reviewers acknowledge the novelty of our work, particularly our use of newly designed metrics to address known issues in previous acquisition strategies, such as oversampling and poor exploration efficiency. To demonstrate the effectiveness of our methods, beyond the theoretical convergence proof, we also include more than ten baselines showing that our approach successfully overcomes the challenges faced by prior methods when optimizing multi-modal objective functions in both single-objective and multi-objective settings.

### **Weaknesses and Questions:**

The below table collects the weaknesses and questions that have been mentioned by at least two reviewers.

| Category | Specific Questions Mentioned | Relevant Reviews |
| :--- | :--- | :--- |
| **Acquisition strategy** | The MFDS strategy description misses some details on the one-step-look-ahead strategies in middle steps and the final greedy strategy. | p3DZ, Lh5F |
| **Test functions** | The baselines are adequate but the test functions are not complex enough. | Lh5F, XTdq |
| **Unclear description** | Lack of clarity that leads to confusion or misunderstanding, including missing ablation studies, absence of averaged results, missing complexity analysis, and unclear distinction between the data used for the surrogate model and for acquisition observations.| XTdq, Lh5F, 9e8V, p3DZ |

For the first concern, we have added additional equations and detailed explanations to clarify how our acquisition strategies are developed. These details are now included in the main text as well as in our responses to reviewers p3DZ and Lh5F.

Regarding the concerns about our test functions, we have added two additional benchmark problems with more complex setups, including more objectives (5) and nontrivial constraints (11). The corresponding results are provided in the main text. We also plan to include further results in future work.

For issues related to unclear descriptions, we have clarified and emphasized information that was already present in the main text or the supplementary materials. For example, we originally included an ablation study by comparing MFDS with its one-step-look-ahead variant MFDS(OSLA), showing the individual contributions of MFDS and the OSLA mechanism. However, we did not explicitly highlight this difference. We thank reviewer XTdq for pointing this out, and we have now made this distinction explicit to avoid confusion. Similarly, in response to reviewer Lh5F’s concern about random seeds, we note that our original experiments already used different random seeds for each trial, but we had not explicitly stated this in the original main text. We appreciate the reviewer’s comment, which helped us make the experimental setup description more precise. Regarding complexity analysis mentioned by reviewer 9e8V, we had included it in the original supplementary materials, and we clarified this in our response. We also addressed the confusion between the data used for training the surrogate model $X$ and the data used for acquisition observations $X\_n$, as raised by reviewers p3DZ and XTdq, and provided a clearer explanation in our responses.

For additional questions raised individually by reviewers, we have addressed each of them in the corresponding responses.

The summary above highlights the main points raised across all reviews and outlines how we have addressed them, which we hope will support the AC in their assessment. We are grateful to all reviewers for their careful reading and valuable suggestions.

---

### Meta-Review · Area_Chair_jCzW · 2026-01-10

**Summary:**

This paper considers the problem of black-box optimization and solves it within the framework of Bayesian optimization (BO). It develops a new aquistion strategy referred to as Minimum Filling Distance Search (MFDS) which selects the next input by minimizing the expected distance between the sequence of inputs queried and the posterior distribution of the optima. It instantiates this principle for both single-objective and multi-objective problems. The paper also provides an asymptotice convergence result for single-objective setting. Experiments demonstrate the efficacy of the proposed acquisition function over some baseline BO acquisition strategies.

The reviewers' appreciated the new intuitive geometric principle, extension to both single and multi-objective settings, and convergence guarantee. They also raised a number of concerns including:
1. acquisition strategy (one-step lookahead in the middle and greedy in the last step)
2. limited experiments in terms of the number of test problems
3. some clarifications on experimental setup, evaluation methodology, and comparison of acquisition strategies

The authors addressed some of these concern satisfactorily (#1) and somewhat satisfactorily (#2 and #3).

I looked at the revised paper and feel that the paper still needs more work on the experimental evaluation to make a stronger case for a new acquisition strategy:
1. The number of evaluations (60 in single-objective and 30 in multi-objective) seem small. It is also not clear why the budget was chosen was different for these two settings.
2. The bar for a new acquisition staretgy is high given the large amount of literature. Specifically, the paper only compares with input space entropy search (PES) and doesn't compare with output space entropy search (MES) and Joint entropy search (JES) which are shown to be improvements over PES in both single and multi-objective settings.
3. One reviewer asked a valid question: "Fig.1 why don't you show comparison between different acquisition functions of the next selection point where the set of all the current points is the same? Current comparison seems to be unfair." The response from the authors' doesn't seem satisfactory.

The AC acknowledges the important contribution made by the paper. For the above three reasons, I recommend rejecting the paper and strongly encourage the authors to improve the paper for future re-submission.

**Reviewer Concerns:**

1. acquisition strategy (one-step lookahead in the middle and greedy in the last step)
2. limited experiments in terms of the number of test problems
3. some clarifications on experimental setup, evaluation methodology, and comparison of acquisition strategies

The authors addressed some of these concern satisfactorily (#1) and somewhat satisfactorily (#2 and #3).

I looked at the revised paper and feel that the paper still needs more work on the experimental evaluation to make a stronger case for a new acquisition strategy:
1. The number of evaluations (60 in single-objective and 30 in multi-objective) seem small. It is also not clear why the budget was chosen was different for these two settings.
2. The bar for a new acquisition staretgy is high given the large amount of literature. Specifically, the paper only compares with input space entropy search (PES) and doesn't compare with output space entropy search (MES) and Joint entropy search (JES) which are shown to be improvements over PES in both single and multi-objective settings.
3. One reviewer asked a valid question: "Fig.1 why don't you show comparison between different acquisition functions of the next selection point where the set of all the current points is the same? Current comparison seems to be unfair." The response from the authors' doesn't seem satisfactory.

**Reviewer Scores:**

Reviewer Lh5F: 4 => 5
Reviewer 9e8V: 6
Reviewer XTdq: 4 => 5 or 6
Reviewer p3DZ: 6

---

### Decision · Program_Chairs · 2026-01-26

Reject